# WEB-COGREASONER: TOWARDS MULTIMODAL KNOWLEDGE-INDUCED COGNITIVE REASONING FOR WEB AGENTS

**Yuhan Guo**[1,2†], **Cong Guo**[1†], **Aiwen Sun**[3], **Hongliang He**[5], **Xinyu Yang**[4], **Yue Lu**[4],
**Yingji Zhang**[7], **Xuntao Guo**[6], **Dong Zhang**[4], **Jianzhuang Liu**[11], **Jiang Duan**[1,12],
**Yijia Xiao**[8‡], **Liangjian Wen**[1*], **Hai-Ming Xu**[9*], **Yong Dai**[10‡]

[1]Southwestern University of Finance and Economics, [2]Shanghai Jiao Tong University,
[3]Central South University, [4]Hithink Research, [5]Westlake University,
[6]Harbin Institute of Technology, [7]University of Manchester
[8]University of California, Los Angeles, [9]University of Adelaide, [10]Fudan University,
[11]Shenzhen Institutes of Advanced Technology, Chinese Academy of Sciences,
[12]Chengdu Everimaging Science and Technology Co., Ltd.

https://Gnonymous.github.io/Web-CogReasoner

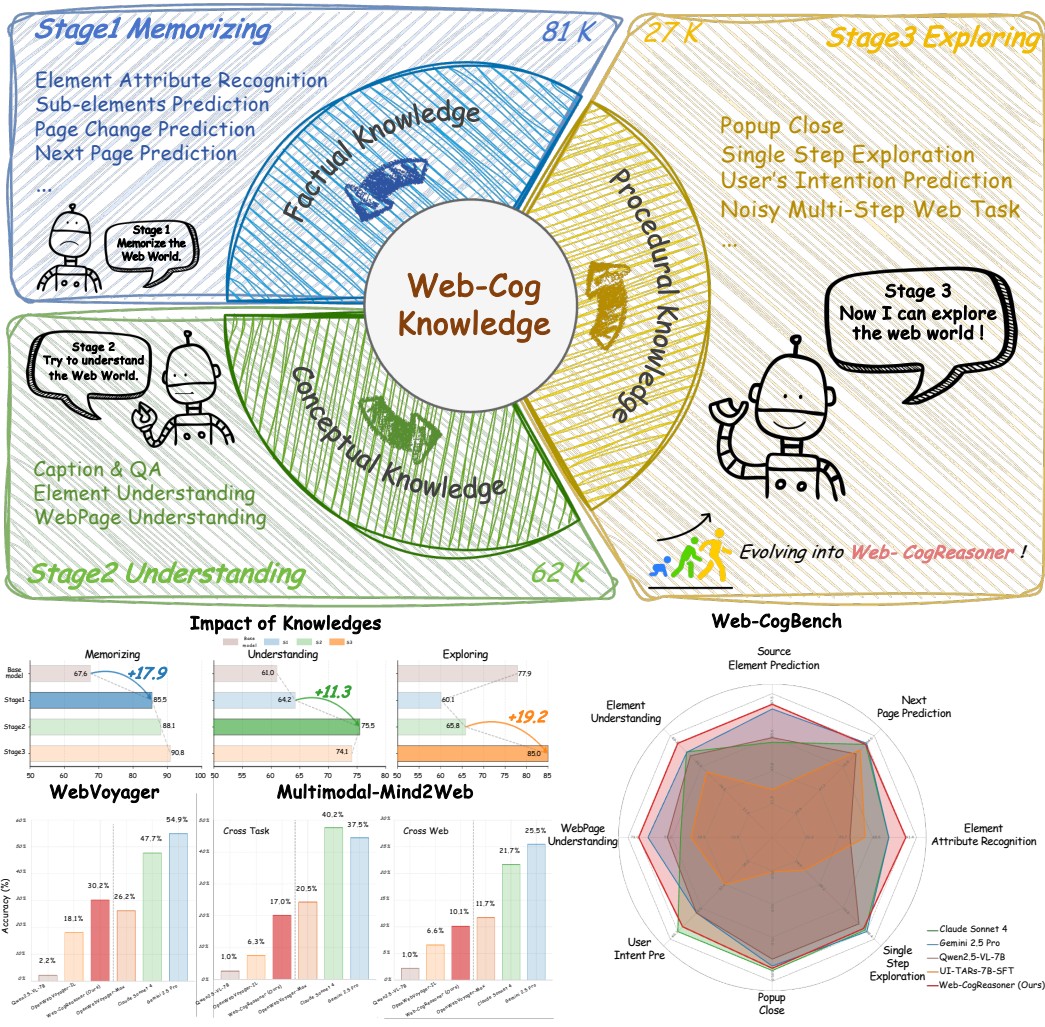

Figure 1: Visualization of the Web-CogKnowledge Framework along with the experimental results.

[†]These authors contributed equally to this work. [‡]Project Leader. [*]Corresponding authors.

## ABSTRACT

Multimodal large-scale models have significantly advanced the development of web agents, enabling them to perceive and interact with the digital environment in a manner analogous to human cognition. In this paper, we argue that web agents must first acquire sufficient knowledge to effectively engage in cognitive reasoning[1]. Therefore, we decompose a web agent's capabilities into two essential phases: **knowledge content learning** and **cognitive processes**. To formalize this, we propose the **Web-CogKnowledge** Framework, which categorizes knowledge into Factual, Conceptual, and Procedural domains. In this framework, knowledge content learning corresponds to the agent's processes of Memorizing and Understanding, which rely on the first two types of knowledge respectively, representing the "what" of learning. Conversely, cognitive processes correspond to Exploring, grounded in Procedural knowledge, defining the "how" of reasoning and action. To facilitate knowledge acquisition, we construct **Web-CogDataset**, a structured resource curated from 14 real-world websites, designed to systematically instill the core knowledge necessary for a web agent. This dataset serves as the agent's conceptual grounding—the "nouns" upon which comprehension is built—as well as the basis for learning how to reason and act. Building on this foundation, we operationalize these processes through a novel knowledge-driven Chain-of-Thought (CoT) reasoning framework, developing and training our proposed multimodal web agent, **Web-CogReasoner**. Extensive experimentation reveals its significant superiority over existing models, particularly in its capacity for generalization to unseen tasks where its structured knowledge proves decisive. To facilitate rigorous and systematic evaluation, we introduce **Web-CogBench**, a comprehensive evaluation suite designed to assess and compare agent performance across the delineated knowledge domains and cognitive capabilities. Our code and data are open sourced at `https://github.com/Gnonymous/Web-CogReasoner`.

## 1 INTRODUCTION

The advent of large-scale models marks a milestone in artificial intelligence, with Large Multimodal Models (LMMs) greatly expanding application horizons. AI agents have become the primary vehicle for deploying these models, enabling capabilities in code generation (Hui et al., 2024; Jiang et al., 2024), image and video synthesis (Huang et al., 2025; Bie et al., 2024; Assran et al., 2025; Liu et al., 2024c), and academic research (Li et al., 2025; Zhang et al., 2025). Recent progress has also highlighted the growing importance of web agents.

Web agents have evolved from early rule-based systems to modern approaches leveraging Large Language Models (LLMs) and Language Vision Models (LVMs) (Wang et al., 2024; Ning et al., 2025; Zhang et al., 2024; Sapkota et al., 2025). LLM-powered agents typically convert HTML or Accessibility Tree inputs into natural language prompts for reasoning and action. With LVMs, agents gain perceptual abilities akin to human vision, allowing them to process multimodal content on web pages. Broadly, web agents can be categorized as: (1) text-only (Zhou et al., 2023; Li et al., 2023), which miss visual cues; (2) vision-only (Qin et al., 2025), which lack structured data; and (3) hybrid (Koh et al., 2024; He et al., 2024b), which integrate both modalities.

LLMs and LVMs pre-trained on general-domain knowledge provide strong foundations but remain limited in specialized tasks, creating performance bottlenecks. Prior knowledge-enhancement methods often lack systematic or theoretical grounding. To address this, we draw inspiration from Bloom's Taxonomy (Ormell, 1974; Conklin, 2005), which divides learning into two phases: Knowledge Content Learning and Cognitive Processes.

In our paradigm, the first phase builds a multi-layered foundation: Factual Knowledge, covering basic concepts, and Conceptual Knowledge, capturing their interrelations. This equips the agent with core web knowledge and its application to familiar tasks. The second phase develops Procedural Knowledge, providing logical frameworks to synthesize prior knowledge for reasoning and exploration. This enables the agent to "learn how to learn," creatively leveraging self-knowledge to

---

[1]Drawing inspiration from Bloom's educational philosophy, a cornerstone of modern pedagogy.

solve novel challenges. This mirrors the human learning trajectory: we first accumulate knowledge through education (Phase 1), and then based on that foundation of knowledge and experience, we learn to apply, innovate, and create (Phase 2).

To support this, we construct Web-CogDataset from 14 prominent websites with 12 fine-grained tasks, and design knowledge-guided reasoning templates combined with imitation learning to instill the required cognitive faculties. Rigorous evaluations on public and in-domain benchmarks show that our method consistently surpasses state-of-the-art baselines, with the performance advantage especially pronounced in knowledge-intensive tasks. These results confirm that structured knowledge acquisition is crucial for enabling agents to excel in complex, domain-specific scenarios.

In summary, our contributions are threefold:

1. Drawing inspiration from Bloom's taxonomy and established human educational paradigms, we propose the Web-CogKnowledge Framework, a systematic, two-phase training methodology designed to enhance the cognitive capabilities of web agents. As shown in Figure 1, built upon this framework, we develop **Web-CogReasoner**. Rigorous benchmarking demonstrates that agents trained under our framework achieve a significant performance improvement over current state-of-the-art models.

2. We construct **Web-CogDataset**, a structured curriculum consisting of 12 fine-grained and progressively challenging tasks. These tasks are meticulously designed to incrementally build the agent's web knowledge, cognitive capability, and higher-order reasoning.

3. To enable comprehensive and robust evaluation, we introduce **Web-CogBench**, a novel benchmark specifically designed to assess whether a web agent possesses the requisite prior knowledge and cognitive capabilities for effective web navigation. This benchmark will be released publicly to foster further research in this area.

## 2 RELATED WORK

### 2.1 WEB AGENT

Early work on web understanding focused on structured HTML, addressing tasks like semantic classification, description generation, and navigation (Gur et al., 2022), with AutoWebGLM (Lai et al., 2024) further applying curriculum learning for structure recognition, component understanding, and progressively complex task execution. More recent studies leverage visual signals: SeeClick (Cheng et al., 2024) linked elements to textual descriptions to enhance localization, CogAgent (Hong et al., 2024) combined high-resolution cross-module modeling with a large GUI dataset for VQA and navigation, OmniParser (Wan et al., 2024) unified text spotting, extraction, and table recognition, UI-TARS (Qin et al., 2025) directly maps screenshots to actions, and UGround (Gou et al., 2024) trained on 10M GUI elements for robust desktop and mobile performance. Multimodal approaches integrate text and vision: WebVoyager (He et al., 2024a) combines screenshots with bounding boxes and accessibility trees, SeeAct (Zheng et al., 2024) grounds text plans via GPT-4V, and TAG (Xu et al., 2025) exploits pretrained attention for grounding without fine-tuning.

### 2.2 WEB AGENT EVALUATION

Benchmarks are categorized into browsing and understanding. For browsing, offline datasets like Mind2Web (Deng et al., 2023), Multimodal-Mind2Web, AutoWebBench (Lai et al., 2024), and WebVLN-v1 (Chen et al., 2024) test multi-step task execution, while online environments such as Mini-WoB++ (Liu et al., 2018), Webshop (Yao et al., 2022), and WebArena (Zhou et al., 2023) allow real-time evaluation. Mini-WoB++ emphasizes low-level UI operations, whereas Webshop and WebArena simulate complex tasks. VisualWebArena (Koh et al., 2024) further adds multimodal inputs for dynamic interactions.

For web understanding, WEBQA (Chang et al., 2022) evaluates open-domain multi-hop reasoning. ScreenQA (Hsiao et al., 2022) and ScreenAI (Baechler et al., 2024) focus on screen comprehension, with ScreenQA targeting UI recognition and contextual QA, while ScreenAI extends this into three subtasks: Screen Annotation, ScreenQA Short, and Complex ScreenQA. Together, they assess layout understanding, semantic interpretation, and reasoning in visually dense interfaces.

# 3 WEB-COGKNOWLEDGE FRAMEWORK

## 3.1 BLOOM'S TAXONOMY

The Bloom's Taxonomy[2] (Anderson & Krathwohl, 2001) presents a two-dimensional framework: knowledge content learning and cognitive processes, embodying a "shallow-to-deep" instructional methodology. It structures learning from foundational facts and concepts to complex procedures, ensuring a solid knowledge base before higher-level reasoning.

This progression can be formalized through four hierarchical types of knowledge: Factual, Conceptual, and Procedural Knowledge. Further details are in Appendix A.1.1.

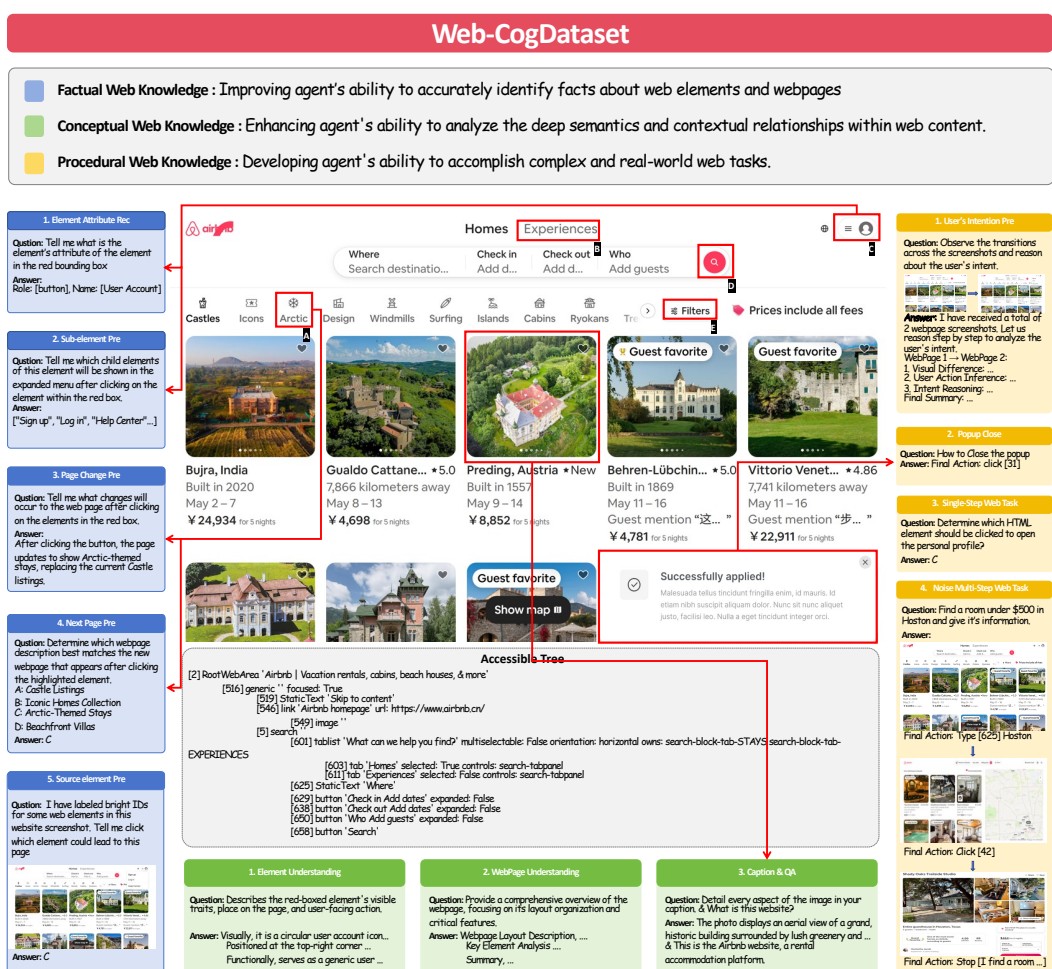

Figure 2: Illustration of Web-CogDataset.

## 3.2 WEB-COGKNOWLEDGE

Motivated by Bloom's taxonomy, we propose a hierarchical web knowledge framework that structures web knowledge according to its taxonomy, collects corresponding knowledge at each level, and trains the model correspondingly. We refer to this knowledge as **Web-CogKnowledge**. This Web-CogKnowledge decomposes knowledge into three levels:

---

[2]https://fctl.ucf.edu/teaching-resources/course-design/blooms-taxonomy/

1. **Factual knowledge**: concrete information extracted from web contents, such as identifying the attributes of individual web elements and predicting the immediate, direct consequences of a single interaction.

2. **Conceptual knowledge**: semantic relationships and abstract patterns underlying webpage content and structures, such as inferring the function of interface components, comprehending the overall purpose and structure of a webpage, and interpreting its multimodal content.

3. **Procedural knowledge**: actionable know-how for accomplishing specific tasks through interaction, including planning, decision-making, and sequential execution. Examples include executing goal-oriented action sequences, inferring user intent from observed behaviors, and handling unexpected interruptions to complete complex tasks.

This taxonomy aligns each knowledge type with a corresponding cognitive competency required for web-based reasoning and interaction.

To fully leverage the potential of **Web-CogKnowledge**, leading to the ultimate realization of the **Web-CogKnowledge Framework**, we first collect multimodal metadata from 14 real-world websites (see Figure 6) and then design **Web-CogDataset** with diverse sets of web tasks for web-agent training in Section 3.3, and finally construct **Web-CogBench** in Section 3.4, built from meticulously chosen subsets. Specifically, we curate the **Web-CogReasoner**'s learning and training process in Section 4 due to its importance and complexity. More details about the data collection and cleaning process can be found in Appendix B.1.

## 3.3 WEB-COGDATASET

By selectively crawling metadata from 14 representative websites and aligning it with the hierarchical design of Web-CogKnowledge, we construct **Web-CogDataset**, a large-scale, hierarchically structured dataset tailored for knowledge-centric web reasoning. The dataset spans three layers of knowledge: Factual, Conceptual, and Procedural. Each is mapped to carefully designed task families that progressively cultivate perception, comprehension, and action-oriented reasoning.

As illustrated in Figure 2, these tasks together form a coherent pipeline that transitions agents from identifying elemental attributes, to grasping semantic patterns and page structures, and finally to executing complex, goal-directed interactions under realistic constraints. This organization mirrors human learning trajectories, ensuring that higher-order reasoning is built on solid perceptual and conceptual foundations.

Detailed task definitions, implementation protocols, and examples are provided in Appendix C.1. Dataset statistics are reported in Table 13. To ensure data quality and diversity, we further analyze the dataset composition and annotation reliability in Appendix C.1.

## 3.4 WEB-COGBENCH

To evaluate the cognitive capabilities enabled by our knowledge-centric framework, we introduce Web-CogBench. While our training dataset is organized by knowledge type (Factual, Conceptual, Procedural), Web-CogBench measures agent performance across three corresponding abilities: Memorizing, Understanding, and Exploring. Curated from a representative subset of Web-CogDataset, it assesses how effectively an agent applies learned knowledge in complex web contexts. Detailed statistics are in Table 1, and the evaluation dimensions align with our hierarchical knowledge framework. A complete definition of all tasks is provided in the appendix C.1.2.

**Memorizing** Assessing the agent's ability to recall and recognize concrete information, directly corresponding to the acquisition of Factual Knowledge. It evaluates whether the agent can accurately identify the attributes of web elements and the state of a webpage.

**Understanding** Measuring the agent's capacity for semantic interpretation, aligning with the mastery of Conceptual Knowledge. It tests whether the agent can move beyond mere identification to comprehend the function of elements and the contextual relationships within a page.

Table 1: Statistics of Web-CogBench.

| Task | Cognition | Metric | #Num |
|------|-----------|--------|------|
| Element Attribute Recognition | | ROUGE-L | 249 |
| Next Page Prediction | Memorizing | Accuracy | 93 |
| Source Element Prediction | | Accuracy | 32 |
| Element Understanding | Understanding | LVM Judge | 200 |
| WebPage Understanding | | LVM Judge | 77 |
| User's Intention Prediction | | LVM Judge | 105 |
| Popup Close | Exploring | Accuracy | 58 |
| Single Step Exploration | | Accuracy | 62 |
| Total | - | - | 876 |

**Exploring** Evaluating the agent's ability to plan and execute goal-oriented actions, reflecting the application of Procedural Knowledge. It assesses whether the agent can strategically navigate the web, handle interruptions, and complete multi-step tasks to fulfill user goals.

## 4 WEB-COGREASONER

### 4.1 PROBLEM SETUP

We model the interaction between Web-CogReasoner and the environment as a partially observable Markov decision process (POMDP): $P = (S, A, O, K, T, R)$, where $S$ denotes the webpage state, $A$ the action space (Table 10), $O$ the observation space, $K$ the internal knowledge, $T$ the transition function, and $R$ the reward function. At each step $t$, the agent receives a screenshot and its accessibility tree (AX Tree), forms a reasoning thought $h_t$, and selects an action $a_t$ under policy $\pi_\theta$. This process continues until task completion or step limits are reached, with binary rewards indicating success or failure.

### 4.2 FRAMEWORK OVERVIEW

Web-CogReasoner is a multimodal knowledge-driven reasoning system built on LVMs and trained on Web-CogDataset (Section 3.3). As illustrated in Figure 3, it tackles complex web tasks by generating a **Knowledge-driven Chain-of-Thought (CoT)**, in which each reasoning stage is explicitly grounded in a layer of Web-CogKnowledge.

**Knowledge-driven CoT Reasoning** The core of Web-CogReasoner is a structured chain-of-thought (CoT) reasoning process (Figure 3), decomposed into three layers: *Factual* (identifying page elements and states: "What is on the page?"), *Conceptual* (inferring roles and interactions: "What does it mean?"), and *Procedural* (planning goal-directed steps: "How to accomplish the task?"). This layered reasoning maps task prompts to executable actions: **Task Prompt** → **Knowledge-driven CoT** → **Plan** → **Action**. The agent initiates with a Task and observes the webpage (Observation Space). This structured reasoning guides the Planning phase, which decomposes the task and formulates a step-by-step strategy. The process culminates in a concrete Action to be executed on the webpage.

## 5 EXPERIMENT

### 5.1 EXPERIMENTAL SETUP

**Models and Baselines** Our proposed **Web-CogReasoner** builds on **Qwen2.5-VL-7B** (Bai et al., 2025), extended with a knowledge-induced reasoning framework. Training details are provided in Appendix D.1. For comparison, we include diverse baselines: the vanilla Qwen2.5-VL-7B

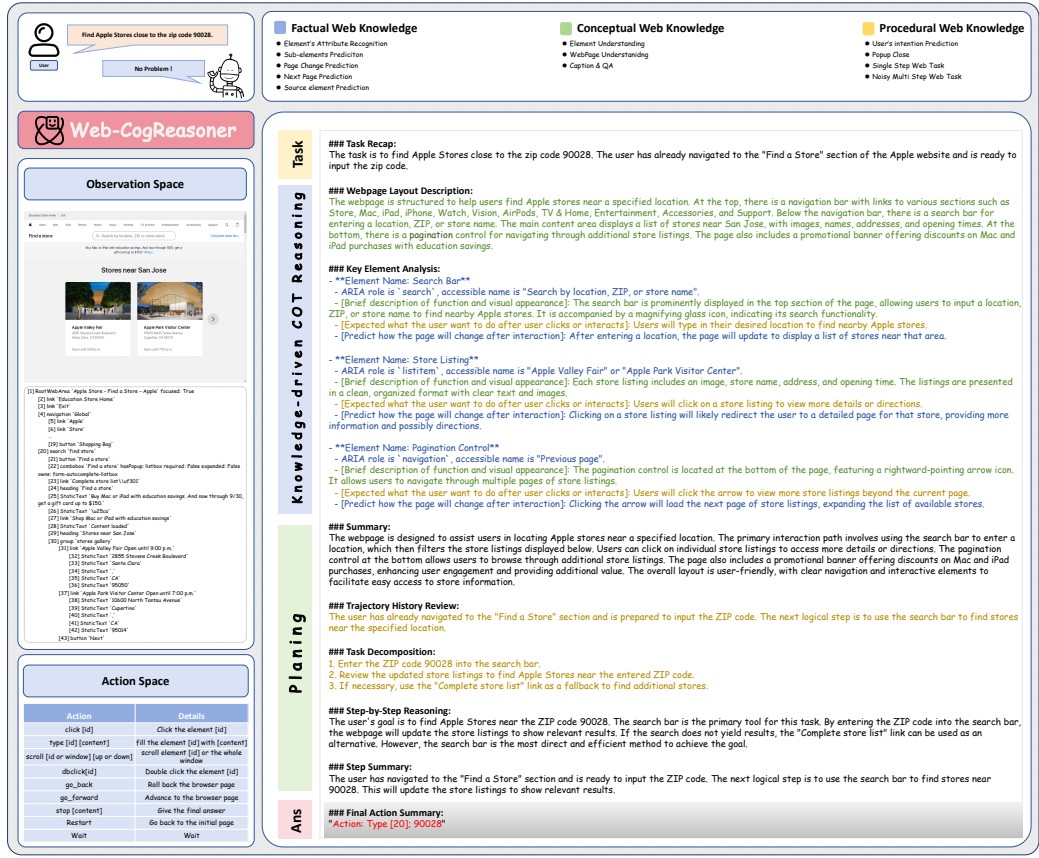

Figure 3: Illustrating the Knowledge-driven Chain of Thought (CoT) process.

(zero/few-shot), **OpenWebVoyager** (He et al., 2024b) in IL and Max variants, the open-source **UI-TARS-7B-SFT** (Qin et al., 2025), and two powerful commercial LVMs, **Claude Sonnet 4** and **Gemini 2.5 Pro**. Together these cover foundational LLMs, end-to-end web agents, and general-purpose multimodal models.

**Datasets**   Web-CogReasoner is trained via supervised fine-tuning on the curated **Web-CogDataset** (Sec. 3.3), which aligns samples with Factual, Conceptual, and Procedural knowledge using screenshots, accessibility trees, and reasoning trajectories. Evaluation is conducted on four datasets. Our custom **Web-CogBench** (Sec. 3.4) assesses cognitive dimensions of Memorizing, Understanding, and Exploring. **VisualWebBench** (Liu et al., 2024b) provides 1.5K curated tasks across 139 real websites, testing grounding and reasoning in diverse settings. **WebVoyager** (He et al., 2024b) contains 643 queries across 15 seen sites, measuring performance in familiar environments. **Online Multimodal-Mind2Web** (Deng et al., 2023) evaluates cross-task and cross-website generalization with queries from both known and unseen domains.

## 5.2   MAIN RESULTS

**Results on Web-CogBench**   We evaluate Web-CogReasoner on **Web-CogBench**, which assesses foundational knowledge and cognitive reasoning across twelve web tasks. As Table 2 shows, our model outperforms both commercial and open-source baselines, owing to the integration of structured Web-CogKnowledge (factual, conceptual, procedural) with Knowledge-driven Chain-of-Thought reasoning. This combination enables accurate perception of web elements and informed, step-wise decision-making. The strong synergy between visual recognition and cognitive reasoning allows Web-CogReasoner to excel on diverse tasks and generalize effectively to web navigation scenarios. We also provide a detailed LVM evaluation protocol and inter-rater reliability analysis in Appendix D.2 to validate the robustness of our scoring metrics.

Table 2: Performance evaluation on the Web-CogBench benchmark.

| Model | Element Attribute Rec | Next Page Pre | Source Element Pre | Element Understanding |
|---|---|---|---|---|
| Claude Sonnet 4 | 79.7 | 93.5 | 62.5 | 62.8 |
| Gemini 2.5 Pro | 79.8 | 94.6 | 84.4 | 62.6 |
| Qwen2.5-VL-7B | 53.2 | 83.9 | 65.6 | 60.0 |
| UI-TARs-7B-SFT | 63.5 | 88.0 | 31.3 | 48.0 |
| **Web-CogReasoner (Ours)** | **91.4** | **93.5** | **87.5** | **69.2** |

| | WebPage Understanding | User Intent Pre | Popup Close | Single Step Exp | Overall |
|---|---|---|---|---|---|
| Claude Sonnet 4 | 54.3 | 64.7 | 100 | 96.8 | 76.8 |
| Gemini 2.5 Pro | 73.5 | 51.9 | 96.6 | 98.4 | 80.2 |
| Qwen2.5-VL-7B | 62.0 | 51.9 | 91.4 | 90.3 | 69.8 |
| UI-TARs-7B-SFT | 48.0 | 32.4 | 25.9 | 33.9 | 46.4 |
| **Web-CogReasoner (Ours)** | **79.0** | **61.4** | **98.3** | **95.2** | **84.4** |

Table 3: Performance evaluation on the VisualWebBench benchmark.

| Model | Perception-Oriented Tasks | | | Perception Avg | Reasoning-Oriented Tasks | | | Reasoning Avg | Overall Avg |
|---|---|---|---|---|---|---|---|---|---|
| | WebQA | HeadOCR | OCR | | Element Ground | Action Prediction | Action Ground | | |
| Claude Sonnet 4 | 73.3 | 72.6 | 96.2 | 80.7 | 81.1 | 96.1 | 96.3 | 91.2 | 85.9 |
| Gemini 2.5 Pro | 74.9 | 70.8 | 95.1 | 80.3 | 91.8 | 96.8 | 90.3 | 93.0 | 86.6 |
| Qwen2.5-VL-7B | 70.8 | 71.7 | 81.4 | 74.6 | 77.5 | 86.8 | 68.0 | 77.4 | 76.0 |
| UI-TARs-7B-SFT | **71.3** | **78.7** | **97.2** | **82.4** | 91.8 | 91.8 | 85.4 | 89.7 | 86.0 |
| **Web-CogReasoner (Ours)** | 67.2 | 72.6 | 97.0 | 79.0 | **96.4** | **96.1** | **88.4** | **93.6** | **86.3** |

**Results on VisualWebBench** We evaluate visual understanding on **VisualWebBench** (Liu et al., 2024b). As Table 3 shows, Web-CogReasoner achieves the highest average score (86.3%), slightly above UI-TARs (86.0%). However, UI-TARs performs poorly on Web-CogBench (46.4%), highlighting that strong visual perception alone does not ensure robust cognitive reasoning. In contrast, Web-CogReasoner consistently excels across both visual and reasoning benchmarks, demonstrating the effective integration of precise visual perception with structured knowledge-driven reasoning — a dual capability essential for reliable web agents.

Table 4: Task success rates on the WebVoyager. The "Overall" score is the average success rate.

| Agent | Allrecipes | Amazon | Apple | ArXiv | GitHub | Booking | ESPN | Coursera |
|---|---|---|---|---|---|---|---|---|
| Claude Sonnet 4 | 26.7% | 87.8% | 48.8% | 69.8% | 68.3% | 2.3% | 45.5% | 83.3% |
| Gemini 2.5 Pro | 60.0% | 63.4% | 62.8% | 67.4% | 68.3% | 9.1% | 56.8% | 73.8% |
| Qwen2.5-VL-7B | 0.0% | 0.0% | 0.0% | 4.7% | 0.0% | 0.0% | 0.0% | 2.3% |
| OpenWebVoyager$_{IL}$ | 17.8% | 12.2% | 20.9% | 14.0% | 14.6% | 9.1% | 9.1% | 31.0% |
| OpenWebVoyager$_{Max}$ | 22.2% | 29.3% | 32.6% | 20.9% | 26.8% | **11.4%** | 11.4% | 42.9% |
| **Web-CogReasoner (Ours)** | **26.7%** | 31.7% | 32.6% | 34.9% | 29.3% | 2.3% | 15.9% | 54.8% |

| | BBC News | Cambridge Dictionary | Google Flights | Google Map | Huggingface | Wolfram Alpha | Overall |
|---|---|---|---|---|---|---|---|
| Claude Sonnet 4 | 23.8% | 37.2% | 4.8% | 80.5% | 48.8% | 82.6% | 47.7% |
| Gemini 2.5 Pro | 52.3% | 76.7% | 4.8% | 75.6% | 58.1% | 82.6% | 54.9% |
| Qwen2.5-VL-7B | 0.0% | 11.6% | 0.0% | 2.4% | 7.0% | 2.2% | 2.2% |
| OpenWebVoyager$_{IL}$ | 9.5% | 37.2% | 9.5% | 22.0% | 20.9% | 26.1% | 18.1% |
| OpenWebVoyager$_{Max}$ | 14.3% | 34.9% | **21.4%** | 29.3% | 32.6% | 37.0% | 26.2% |
| **Web-CogReasoner (Ours)** | **14.3%** | **55.8%** | 9.5% | **39.0%** | **37.2%** | **39.1%** | **30.2%** |

Table 5: Performance comparison on the Online-Mind2Web under cross-task and cross-website

| Agent | Cross-task (Unseen Tasks) | | | | Cross-web (Unseen Websites) | | | |
|---|---|---|---|---|---|---|---|---|
| | Entertainment | Shopping | Travel | Overall | Entertainment | Shopping | Travel | Overall |
| Claude Sonnet 4 | 44.9% | 35.3% | 40.0% | 40.2% | 45.5% | 6.7% | 14.0% | 21.7% |
| Gemini 2.5 Pro | 46.9% | 35.3% | 28.3% | 37.5% | 42.4% | 10.0% | 23.3% | 25.5% |
| OpenWebVoyager$_{Max}$ | 22.4% | 29.4% | 15.2% | 20.5% | 3.0% | 8.7% | 23.3% | 11.7% |
| Qwen2.5-VL-7B | 2.2% | 0.0% | 0.0% | 1.0% | 3.0% | 0.0% | 0.0% | 1.0% |
| OpenWebVoyager$_{IL}$ | 8.2% | 5.9% | 4.3% | 6.3% | 3.0% | 5.8% | 4.7% | 6.6% |
| **Web-CogReasoner (Ours)** | **16.3%** | **23.5%** | **15.2%** | **17.0%** | **12.1%** | **7.7%** | **9.3%** | **10.1%** |

**Results on Online Web Tasks** We evaluate Web-CogReasoner on live web tasks using **WebVoyager** and **Online Multimodal-Mind2Web** to assess practical utility and generalization to unseen websites and multi-step tasks (UI-TARs-7B-SFT omitted due to missing online inference scripts). Web-CogReasoner achieves state-of-the-art results among open-source agents, suggesting that integrating structured Web-CogKnowledge with Chain-of-Thought reasoning supports more reliable perception and execution. For Mind2Web, OpenWebVoyagerMax is not a strictly comparable zero-shot baseline due to additional sample collection and retraining on high-error sites; nevertheless, without task-specific fine-tuning, Web-CogReasoner outperforms OpenWebVoyagerIL and stays competitive with OpenWebVoyager$_{Max}$, highlighting robust generalization.

Table 6: Average steps per successful task across different benchmarks.

| Agent | WebVoyager | Mind2Web Cross-Task | Mind2Web Cross-Web | Final Avg |
|---|---|---|---|---|
| Claude | 7.35 | 10.89 | 11.04 | 9.76 |
| Gemini | 6.68 | 7.74 | 10.30 | 8.24 |
| OpenWebVoyager$_{Max}$ | 5.07 | 7.59 | 6.91 | 6.52 |
| Qwen2.5-VL-7B | 7.69 | 12.00 | 13.00 | 10.9 |
| OpenWebVoyager$_{IL}$ | 5.26 | **7.00** | 9.29 | 7.18 |
| **Ours** | **4.73** | 7.37 | **8.89** | **7.00** |

**Results on Average Steps** Table 6 reports the average number of steps for successful online tasks. Our approach consistently achieves high efficiency, particularly in cross-domain scenarios, indicating that the model effectively balances streamlined task execution with robust generalization to unseen environments.

## 5.3 ABLATION STUDY

To empirically validate the effectiveness of our Web-CogKnowledge Framework, we conduct a two-fold ablation study. First, we evaluate the cumulative gains of our curriculum learning strategy (Table 7). Second, to address specific inquiries regarding the necessity of each knowledge layer and the reasoning mechanism, we provide a detailed component analysis on both Web-CogBench and WebVoyager (Tables 8 and 9).

Table 7: Cumulative Gains: Impact of progressive knowledge training on Web-CogBench.

| Model Configuration | Memorizing | Understanding | Exploring | Overall |
|---|---|---|---|---|
| Qwen2.5-VL-7B (Base Model) | 67.6 | 61.0 | 77.9 | 69.8 |
| + Factual Knowledge (S1) | **85.5** (+17.9) | 64.2 | 60.1 | 72.1 |
| + Conceptual Knowledge (S2) | 88.1 | **75.5** (+11.3) | 65.8 | 78.3 |
| + Procedural Knowledge (S3) | 90.8 | 74.1 | **85.0** (+19.2) | **84.4** |

**Cumulative Impact of Curriculum Learning** We ablate the curriculum components by incrementally adding Factual, Conceptual, and Procedural knowledge to the base Qwen2.5-VL-7B and evaluating on Web-CogBench. Factual knowledge improves perceptual grounding for element

recognition (Memorizing), Conceptual knowledge adds semantic/functional understanding (Understanding), and Procedural knowledge enables goal-directed planning and execution (Exploring). Additional qualitative examples are provided in Appendix E.

Table 8: Component Analysis on Web-CogBench: Validating hierarchical dependency.

| Model | Memorizing | Understanding | Exploring | Overall |
|---|---|---|---|---|
| Base model | 67.6 | 61.0 | 77.9 | 69.81 |
| S1 only | 85.5 | 64.2 | 60.1 | 72.12 |
| S2 only | 59.88 | 68.03 | 60.00 | 61.96 |
| S3 only | 52.82 | 46.40 | 78.00 | 60.66 |
| S1+S2 | 88.1 | **75.5** | 65.8 | 78.33 |
| S1+S3 | 85.11 | 53.53 | 82.31 | 76.17 |
| S2+S3 | 64.87 | 69.74 | 81.41 | 72.29 |
| **S1+S2+S3 (Full)** | **90.8** | 74.1 | **85.0** | **84.44** |

Table 9: Impact of Knowledge & KCoT on WebVoyager, real-world online tasks.

| Model | Amazon | Cambridge | Coursera | GitHub | Overall |
|---|---|---|---|---|---|
| S1 only | 12.19% | 25.58% | 14.28% | 7.14% | 12.67% |
| S3 only | 17.07% | 11.62% | 16.66% | 14.28% | 13.14% |
| S1+S3 | 29.26% | 34.88% | 28.57% | 16.66% | 23.47% |
| S1+S2+S3 (w/o KCoT) | 19.51% | 51.16% | 26.19% | 23.80% | 25.35% |
| **S1+S2+S3 (w/ KCoT)** | **31.7%** | **55.8%** | **54.8%** | **29.3%** | **42.9%** |

**Hierarchical Dependency of Knowledge**    To verify that our curriculum stages are hierarchically dependent layers, we analyze the component ablations in Table 8 and the online results in Table 9.

- **Low-level Knowledge is a Prerequisite:** In Table 8, single-stage models (S2 only, S3 only) underperform on overall metrics. Adding Factual training (S1) substantially strengthens higher stages; for instance, on WebVoyager (Table 9), S1+S3 nearly doubles the success rate vs. S3 only (**23.47% vs. 13.14%**), indicating that procedural exploration relies on accurate factual grounding.

- **Integration is Critical:** Although individual stages specialize (e.g., S3 benefits Exploring), only the integrated model (S1+S2+S3) achieves consistently strong performance across dimensions, suggesting that effective web agents require the full cognitive stack.

**Reasoning Activation via KCoT**    Finally, we investigate the role of our reasoning framework. While the full combination of data (S1+S2+S3) builds a strong latent representation, explicit reasoning is required to utilize it. As shown in Table 9, removing the Knowledge-driven Chain-of-Thought (w/o KCoT) causes a sharp drop in online success rate from **42.9% to 25.35%**. This indicates that KCoT acts as a crucial activator, bridging the gap between *possessing* knowledge and *applying* it dynamically for decision-making.

## 6    CONCLUSION

We present Web-CogReasoner, a multimodal cognitive-inspired framework for web agents that systematically instills Factual, Conceptual, and Procedural Knowledge, following Bloom's Taxonomy. By coupling the Web-CogKnowledge Framework with the Web-CogDataset and Web-CogBench, our approach enables interpretable, step-wise reasoning through knowledge-driven Chain-of-Thoughts, yielding strong performance on complex web navigation and instruction-following tasks. Ablation and qualitative analyses confirm the indispensability of each knowledge stage, demonstrating how a structured curriculum produces robust perceptual and cognitive capabilities. While current results rely on imitation learning, future work aims to integrate reinforcement learning to enhance exploration, generalization, and autonomous discovery of procedural knowledge, advancing toward truly adaptive and self-directed web agents.

## 7 ACKNOWLEDGMENTS

This work was supported by the Major Science and Technology Special Project of the Sichuan Provincial Department of Science and Technology (Grant No. 2024ZDZX0002), the Sichuan Provincial Innovation Group Project (Grant No. 2024NSFTD0054), Fundamental Research Funds for the Central Universities (JBK202511081).

## 8 ETHICS STATEMENT

This work does not involve human subjects, private or sensitive data, or any personally identifiable information. All datasets employed in our experiments are publicly available and have been used in accordance with their respective licenses. We acknowledge the potential societal risks associated with large language models, including issues of fairness, bias, and misuse. While these concerns are not the primary focus of this work, we have taken steps to ensure responsible experimentation, including transparent reporting of datasets, models, and evaluation protocols. The release of our code and models will be accompanied by appropriate documentation and usage guidelines to mitigate unintended applications.

## 9 REPRODUCIBILITY STATEMENT

We have made every effort to ensure that our results are fully reproducible. All relevant details—including the proposed methodology, model architecture, training objectives, evaluation protocols, hyperparameter settings, ablation configurations, and data preprocessing pipeline—are thoroughly documented in the appendix. All datasets used in our experiments are publicly accessible, and instructions for reproducing our experiments are clearly provided. To further support independent verification and extension of our work, we will release the source code, trained model checkpoints, and experiment scripts in the near future.

## 10 LLM USAGE

Large Language Models (LLMs) were used to aid in the writing and polishing of the manuscript. Specifically, we used an LLM to assist in refining the language, improving readability, and ensuring clarity in various sections of the paper, The model helped with tasks such as sentence rephrasing, grammar checking, and enhancing the overall flow of the text.

In our research, LLMs were used to assist in generating training tasks. Specifically, in Section 13, we used the LLM infer the functions of web page elements based on web page screenshots and structured text. We then used these results to build training data, which helped improve the target model's understanding and prediction capabilities.

It is important to note that the LLM was not involved in developing the research concepts, designing the research methodology, or formulating the experimental protocols. All research concepts, ideas, and analyses were independently developed and implemented by the authors. The authors bear full responsibility for the content of the manuscript, including any text generated or polished by the LLM. We have ensured that the text generated by the LLM complies with ethical guidelines and does not involve plagiarism or scientific misconduct.

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

## A  COGNITIVE FRAMEWORK AND INTERACTION ENVIRONMENT

### A.1  BLOOM'S TAXONOMY AND COGNITIVE STAGES

#### A.1.1  BLOOM'S TAXONOMY

1. **Factual Knowledge:** The foundational layer, encompassing the basic, discrete elements of a discipline that a student must know, such as essential terminology and specific, isolated details.

2. **Conceptual Knowledge:** The synthesis of factual elements into a coherent, organized structure. This level focuses on the interrelationships between basic elements, including knowledge of classifications, principles, generalizations, theories, and models.

3. **Procedural Knowledge:** The knowledge of how to perform a task or inquiry. This involves an understanding of specific skills, algorithms, techniques, and methods, representing a shift from "knowing-what" to "knowing-how."

#### A.1.2  STAGES OF HUMAN KNOWLEDGE AND COGNITIVE DEVELOPMENT

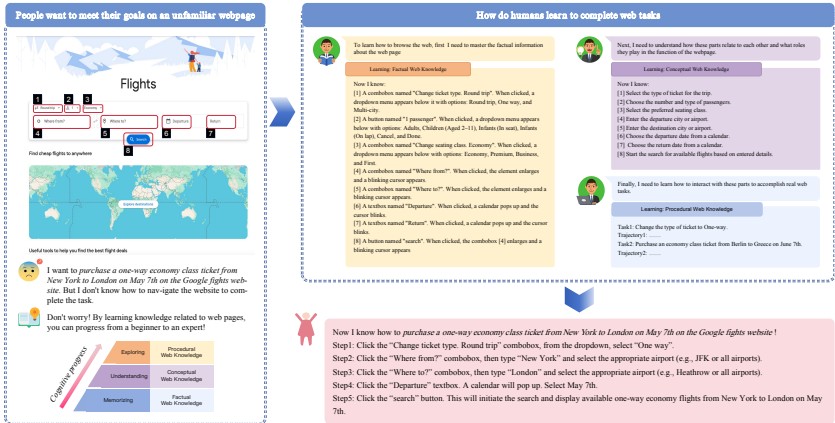

Figure 4: How people handle unfamiliar web pages. People learn factual, conceptual, and procedural knowledge to memorize, understand, and explore the web, ultimately completing specific tasks.

Cognitive science typically classifies human knowledge into three categories—factual, conceptual, and procedural—which correspond to different stages of cognitive development: perceiving, understanding, and executing. This taxonomy captures the natural trajectory of human learning: starting with the perception of concrete facts and data (factual knowledge), progressing toward abstract comprehension of concepts and relationships (conceptual knowledge), and ultimately acquiring the ability to carry out complex, goal-oriented behaviors through practiced routines and strategies (procedural knowledge). An illustrative example is shown in Figure 4.

### A.2  WEB INTERACTION ACTION SPACE

Table 10 summarizes the action space of Web-CogReasoner for web interaction tasks. The table categorizes actions into two groups: (1) element-specific operations, such as clicking, typing, double-clicking, and scrolling individual elements; and (2) page-level control actions, including navigation commands, task restart, waiting, and final answer submission. This structured action space enables the agent to perform a diverse set of interactions, facilitating comprehensive exploration and manipulation of web pages in a controlled and systematic manner.

Table 10: Action space of Web-CogReasoner for web interaction.

| Instruction | Description |
|---|---|
| click [id] | Click an element |
| type [id] [content] | Input specified content into an element |
| scroll [id or WINDOW] [up/down] | Scroll an element or the page up/down |
| dbclick [id] | Double-click an element |
| go_back | Navigate to the previous webpage |
| go_forward | Navigate to the next webpage |
| stop [content] | Submit the final answer |
| Restart | Restart the current task |
| Wait | Wait for one second before proceeding |

## B  WEB-COGDATASET: CONSTRUCTION AND DATA QUALITY

### B.1  DATA SOURCING AND METADATA COLLECTION

To collect comprehensive metadata from web pages, we developed a data collection tool based on Playwright. This tool performs deep traversal and interaction by systematically clicking on all elements within each page. We define each round of interaction (i.e., one click) as a layer, and using this iterative approach, we collected Layer 1 to Layer 6 data from 14 different websites (for complete website information, refer to Table 6 )

Table 11: Web elements's meta-data.

| Data | Description |
|---|---|
| css | Element's CSS selectors |
| allcss | CSS selector sequence of preceding elements |
| outerhtml | element's outerhtml |
| location | element's boundingbox |
| role | element's role |
| name | element's name |

For each web element, we capture its standalone screenshot, as well as screenshots taken before clicking (both with and without a red bounding box), after hovering, and after clicking. See Figure 5 for an example. We also collect the following metadata: CSS, allCSS, outerHTML, and location. Additionally, we extract semantic information from each element based on its outerHTML. If a role attribute is explicitly defined, we use its value directly as the element's semantic role. Otherwise, we infer the role by mapping the tag name using the WAI-ARIA specification. Similarly, to determine the element's semantic name, we extract the value of the aria-label if present; otherwise, we get its textual content. See Table 11 for detailed metadata of the web elements.

After collecting both the visual and semantic metadata, we present the corresponding screenshots of each clickable element to Qwen2.5-VL 72B. The model is instructed to:

- analyze the visual changes on the page after hovering over and clicking the target element;

- identify and list any sub-elements that appear upon interaction (e.g., when a dropdown menu is triggered by clicking);

- infer and generalize the functional purpose of the element.

For the functional purpose prediction, the model is additionally required to provide a confidence score. If this score remains below 0.5 after three retries, the prediction is excluded from evaluation.

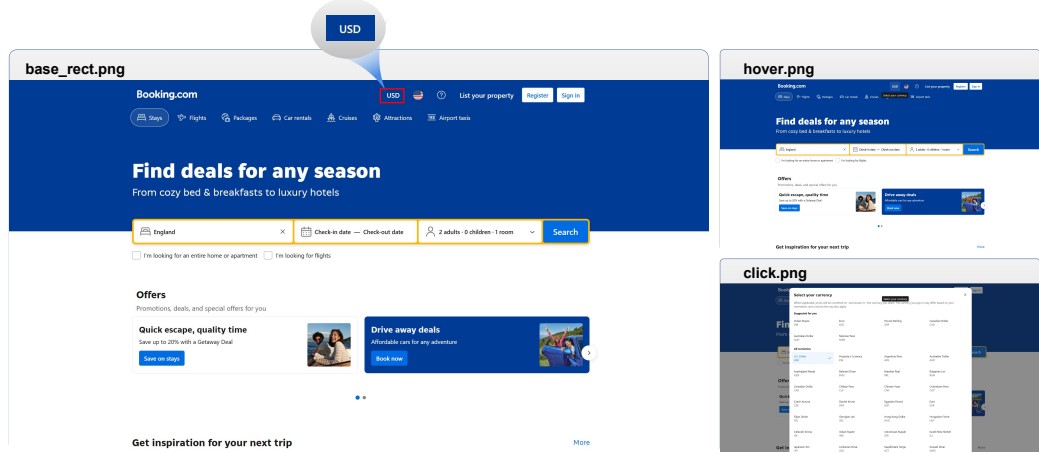

Figure 5: Example visual states of a web element ("USD") we captured. Shown are: the element highlighted in the full-page view (base_rect.png), the hover state (hover.png), and the click state (click.png). These screenshots illustrate how the element's visual context evolves through user interactions.

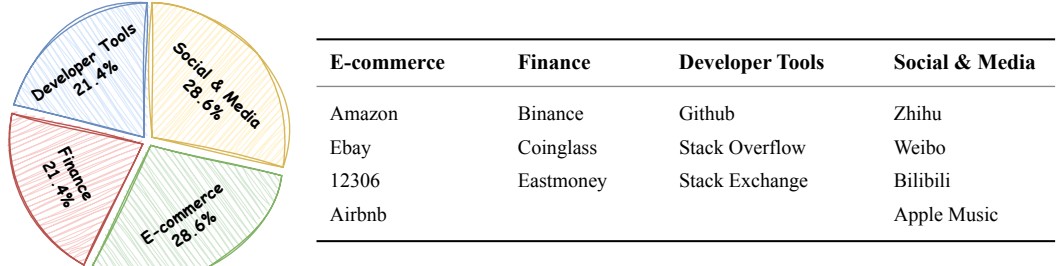

| E-commerce | Finance | Developer Tools | Social & Media |
|------------|---------|-----------------|----------------|
| Amazon | Binance | Github | Zhihu |
| Ebay | Coinglass | Stack Overflow | Weibo |
| 12306 | Eastmoney | Stack Exchange | Bilibili |
| Airbnb | | | Apple Music |

Figure 6: Statistics of selected websites by category.

## B.2 DATA COMPOSITION AND DOMAIN BALANCE

To avoid domain overfitting and ensure both interaction depth and broad generalization, Web-CogDataset employs a strategic hybrid data composition:

1. **Depth via Self-Collected Data:** We selected 14 complex websites for high-depth interaction mining to capture intricate logic often missed by general crawls. As shown in Figure 6, we strictly maintained category balance across E-commerce, Finance, Developer Tools, and Social Media within this subset to prevent bias toward any single domain.

2. **Breadth via Open-Source Augmentation:** To address the concern of limited domain diversity (e.g., lack of News, Education, or Forums), we incorporated and enhanced large-scale open-source datasets, including **MultiUI** (Liu et al., 2024a), **Mind2Web** (Deng et al., 2023), and **OpenWebVoyager** (He et al., 2024b). Notably, MultiUI is derived from FineWeb (Common Crawl), providing massive coverage of general-purpose webpages. This combination ensures our model generalizes to the "wild" web and is not overfitted to specific interaction styles like financial trading or shopping.

## B.3 ANNOTATION RELIABILITY

We validated our automated annotations via double-blind human verification and cross-model consistency checks (e.g., using GPT-4o). As shown in Table 12, the error rate is minimal.

Table 12: Reliability Check of Web-CogDataset Annotations.

| Annotation Task | Human Verification (Acc) | Cross-Model Consistency |
|---|---|---|
| Element Attribute | 99.2% | 98.5% |
| Page Change Pred | 97.5% | 96.8% |
| Sub-element Pred | 96.8% | 95.4% |
| **Average** | **97.8%** | **96.9%** |

## C DETAILED TASK SPECIFICATIONS AND STATISTICS

### C.1 OVERALL DATASET STATISTICS

#### C.1.1 WEB-COGDATASET

Table 13: Statistics of Web-CogDataset.

| Knowledge | Task | Statistics | |
|---|---|---|---|
| | | Subtotal | Total |
| Factual Web Knowledge | Element Attribute Recognition | 37K | |
| | Sub-element Prediction | 1K | |
| | Page Change Prediction | 24K | 81K |
| | Next Page Prediction | 18K | |
| | Source Element Prediction | 1K | |
| Conceptual Web Knowledge | Element Understanding | 40K | |
| | WebPage Understanding | 7K | 62K |
| | Caption & QA | 15K | |
| Procedural Web Knowledge | User's Intention Prediction | 2K | |
| | Popup Close | 1K | |
| | Single-Step Web Task | 17K | 27K |
| | Noisy Multi-Step Web Task | 7K | |

#### C.1.2 WEB-COGBENCH

- **Element Attribute Recognition**: Given a screenshot with a highlighted interactive element, the model predicts its semantic role (e.g., button, link) and accessible name (e.g., "Submit", "Search"), relying solely on visual cues.

- **Next Page Prediction**: The model predicts the subsequent page that results from interacting with a specific element on the current page. To enhance generalization, we design two types of tasks: multiple-choice questions and open-ended responses.

- **Source Element Prediction**: Given two screenshots, the current page and the resulting target page, then the model identifies which of the visually marked elements on the current page leads to the target, simulating visual cause-and-effect reasoning.

- **Element Understanding**: For a specific interactive element, the model generates an open-ended paragraph that comprehensively describes the element's Visible Traits (e.g., text, shape, styling), its On-page Location (e.g., header, sidebar, main content), and its likely User-facing Function (e.g., playing a video, navigating to a new page), relying solely on visual context.

- **WebPage Understanding**: Given a full-page screenshot, the model generates a comprehensive overview describing the webpage's Layout Organization (e.g., header, event information, seating chart, filter panel), Key Element Analysis (e.g., element attribute, description, function, interaction, expected outcome), and a Summary of the WebPage. This enables a thorough understanding of the webpage's structure and functionality.

- **User's Intention Prediction**: The model infers high-level user intent from a sequence of webpage screenshots representing an interaction trajectory, requiring visual understanding and temporal reasoning. The task is built on the MultiModal-Mind2Web (Deng et al., 2023) dataset, mapping screenshot sequences to natural language instructions.

- **Popup Close**: The model identifies and dismisses popups (e.g., notification modals, login forms) on synthesized webpage screenshots, using a dataset of 51 popup components from JS Design[3] overlaid on OpenWebVoyager (He et al., 2024b) webpages, with combinatorial augmentation of closing strategies for diverse training.

- **Single Step Exploration**: This task is derived from a multi-step trajectory exploration task and has been decomposed into single-step exploration subtasks. For each step, the model receives as input the corresponding accessibility tree (AxTree) and a screenshot of the current webpage. Based on these observations, the model performs reasoning to generate the appropriate action along with the target object, effectively simulating a realistic single-step web navigation scenario. By breaking down complex multi-step interactions into atomic actions, this setup allows for fine-grained evaluation of the agent's decision-making capabilities and supports systematic analysis of its performance in web exploration tasks.

Table 13 summarizes the overall statistics of Web-CogDataset. The dataset covers three layers of knowledge—Factual, Conceptual, and Procedural— each corresponding to distinct families of web reasoning tasks. Factual Web Knowledge focuses on recognizing attributes, predicting element relationships, and modeling page transitions, totaling 81K instances. Conceptual Web Knowledge emphasizes semantic understanding and cross-element comprehension, with 62K instances. Procedural Web Knowledge involves action-oriented reasoning tasks, such as predicting user intentions and executing goal-directed interactions, comprising 27K instances. Together, these task distributions reflect the hierarchical design of Web-CogDataset and ensure balanced coverage from low-level perception to high-level reasoning.

## C.2 FACTUAL WEB KNOWLEDGE TASKS

**Element Attribute Recognition**   We define the Element Attribute Recognition task to assess a model's capability to infer the interactive semantics of web elements exclusively from visual input. Given a full-page screenshot with a specific interactive element marked by a red bounding box, the model is tasked with predicting two key attributes:

- the semantic role of the highlighted element (e.g., "button", "link", "checkbox"),

- the semantic name, which refers to the element's accessible textual description (e.g., "Submit", "Search", "Next").

The ground truth for both attributes is derived from the element-level metadata collected as described in Section B.1. This task simulates the human cognitive ability to interpret the function of web interface components through visual perception alone, without relying on HTML structure or programmatic representations.

**Sub-elements Prediction**   We define the Element Sub-element Prediction task to evaluate a model's ability to infer the hierarchical structure of web interfaces—specifically, to identify the sub-elements that become visible upon interaction with a given parent element, using only visual information. In each task instance, the model is presented with a full-page screenshot in which a specific interactive element is highlighted by a red bounding box. The model is instructed to predict the set of sub-elements (e.g., menu items, dropdown options) that appear as a direct result of interacting with the highlighted element, such as clicking or hovering. The ground truth annotations for sub-elements are derived from the element-level metadata collected during the dynamic interaction process, as detailed in SectionB.1. This task simulates the human cognitive process of understanding interactive dependencies in a graphical interface—recognizing not only that a component is clickable, but also predicting its dynamic expansion behavior.

---

[3]https://js.design/.

**Page Change Prediction**  We define the Page Change Prediction task to evaluate a model's capability to infer the visual consequences of interacting with a specific web element, relying solely on visual input. In this task, the model is presented with a full-page screenshot in which a target interactive element is highlighted by a red bounding box. The model is required to prediet, in an open-ended format, the visual changes that are likely to occur on the page after the element is clicked. The ground truth for this task is obtained from the generated responses of Qwen-VL-72B, which were produced based on visual metadata, as detailed in SectionB.1. This task is designed to simulate the human cognitive ability to anticipate the dynamic behavior of web interfaces through perception alone—without access to the underlying source code or prior knowledge of the page logic.

**Next Page Prediction**  We define the Next Page Prediction task to evaluate a model's ability to forecast navigation outcomes. Given a full-page screenshot with a highlighted interactive element, the model must predict the subsequent page that would result from interacting with that element. To ensure generalization capability, we implement two evaluation formats: multiple-choice selection (choosing from 4-5 possible next pages) and open-ended generation (describing the expected next page). Ground truth is derived from actual navigation sequences recorded during web interactions. This task assesses the agent's understanding of functional relationships between interface elements and destination pages.

**Source Element Prediction**  We define the Source Element Prediction task to assess a model's ability to identify which element on a webpage leads to a specific target page, using only visual input. The model is given two screenshots: one showing the current webpage with 4-10 candidate elements marked by bounding boxes, and another showing the resulting target page. Based on visual cues alone, the model should determine which candidate element would trigger the transition to the target page when interacted with. This task simulates the human ability to reason about visual cause-and-effect relationships in web navigation, without relying on code or prior knowledge of page logic.

## C.3    Conceptual Web Knowledge Tasks

**Element Understanding**  This task requires the model to produce a comprehensive, open-ended description of a highlighted element's visual appearance, functional semantics, and placement on the webpage. Specifically, the output should cover: (1) Visual Traits (text, shape, iconography); (2) Location (e.g., `top-right`, `footer`); and (3) Function (e.g., `navigates to user profile`). This task simulates abstract comprehension from concrete element appearance.

**WebPage Understanding**  In this task, the model must generate a detailed and structured overview of the entire page. The response includes layout segmentation (e.g., header, sidebar, content area), key modules (e.g., search panel, product gallery), and a summary of page purpose and interactivity. This facilitates understanding of page-wide structure and intent.

**Caption & QA**  We define the Caption & QA task to evaluate a model's capability to comprehend and reason over both image and non-image content embedded within webpages. This task comprises four subtasks:

- **Embedded Image Captioning:** Given a full-page screenshot containing one or more embedded images, the model is required to generate a detailed and semantically meaningful caption for each image, describing its visual content and its contextual relevance within the surrounding webpage layout.
- **Embedded Image QA:** Given a question grounded in the content of an embedded image within a webpage screenshot, the model must produce an accurate, context-aware answer using only visual information. These questions may refer to image content (e.g., "What brand is shown in the ad?") or its function in the UI.
- **Webpage Captioning:** The model is tasked with generating an open-ended description of the webpage's content, layout, and interactive purpose, treating the entire screenshot as input. The generated caption should reflect both structural composition and the inferred user intent of the webpage.

- **Webpage QA:** Given a full-page screenshot and a natural language question referring to any aspect of the page (e.g., title, layout, purpose, textual content), the model must generate a grounded and precise answer based on visual and spatial information.

All four subtasks are derived from the Multi-UI (Liu et al., 2024a) dataset, which provides rich annotations for webpage visual elements and user-facing semantics. Together, these subtasks measure a model's ability to perform grounded visual-language understanding at both local (element-level) and global (page-level) scales.

### C.4 PROCEDURAL WEB KNOWLEDGE TASKS

**User's Intention Prediction**   Built on the MultiModal-Mind2Web dataset—which provides natural language instructions, action trajectories, and aligned web page screenshots—we introduce a novel multi-modal task: inferring the user's high-level intent from a sequence of visual observations. Unlike traditional imitation learning or instruction-following tasks, our setting requires the model to infer why a trajectory occurred, rather than how to execute it. Solving this task demands both visual understanding and temporal reasoning. The details of this task are as follows:

1. Task Definition: Given a sequence of web page screenshots $p_1, p_2, \ldots, p_n$ representing a user's interaction trajectory, the objective is to predict the original user instruction y that guided the sequence. Each screenshot $p_t$ corresponds to the visual observation at step $t$ of a successful task execution. Formally, the model learns a mapping: $f : \{p_1, p_2, \ldots, p_n\} \rightarrow Q$ where $Q$ is the natural language instruction.

2. Dataset Construction: We construct our dataset by processing the original MultiModal-Mind2Web corpus. For each task, we extract only the visual observations—i.e., the sequence of web page screenshots corresponding to each step in the execution trajectory. We then pair each screenshot sequence with the original natural language instruction as the supervision signal.

**Popup Close**   We curated a collection of 51 popup components from JS Design website, encompassing a diverse range of visual styles and functional categories, such as notification modals, alert dialogs, and login forms. This diversity ensures comprehensive coverage of real-world popup use cases. For background webpages, we utilized the OpenWebVoyager (He et al., 2024b) dataset, which contains a large number of authentic webpage screenshots with varied layouts and content, providing a rich foundation for synthesizing realistic popup-injected webpages. To construct the training data for this task, we employed the following procedure:

1. Synthesizing Webpage Screenshots with Popups: We randomly overlaid popup images onto background webpage screenshots to simulate webpages containing popups. During synthesis, we introduced variability by randomly adjusting the popup's size and position and modifying the brightness and sharpness of the background images, thereby enhancing visual diversity and realism.

2. Generating Popup AX Tree: Each popup image was processed using Qwen-VL-2.5-32B to generate an ARIA-compliant AX Tree. To simulate diverse structural configurations, we randomly modified the index values of popup AX Tree elements and inserted the popup AX Tree into different locations within the original webpage's AX Tree, resulting in a combined AX Tree that reflects realistic variations in webpage structure.

3. Generating Popup Closing Strategies: We then instructed Qwen-VL-2.5-32B to identify all n possible methods for closing the popup, based on the popup image and its corresponding AX Tree. Recognizing that, in practical settings, any correct method is sufficient, we applied combinatorial augmentation to the n methods. Specifically, we enumerated all non-empty subsets of the n strategies, yielding a total of $2^n - 1$ distinct answer combinations. This expansion significantly broadens the training distribution and increases the model's exposure to diverse correct solutions.

4. Constructing the Training Dataset: Using the synthesized webpage screenshots and the enriched AX Tree, we constructed a dataset for training models on popup dismissal. Each data point comprises:

- Input: a webpage screenshot with an embedded popup and the corresponding AX Tree;

- Output: valid methods for closing the popup.

**Single-Step Web Task**    We define the Single-Step Web Task to evaluate a model's ability to ground high-level user intentions in visual webpage elements. Each task instance includes a full-page screenshot from a real-world webpage, a concise natural language instruction (e.g., "Search for a product", "Log into the system"), and several candidate elements marked by red bounding boxes.

The model must identify which element, if clicked, would successfully fulfill the given task. This setup simulates perceptual grounding of user intent—matching natural language goals to actionable UI targets based solely on visual cues.

All samples are directly sourced from the Multi-UI (Liu et al., 2024a) dataset, which provides rich, annotated webpage screenshots paired with task descriptions and labeled ground-truth targets. No trajectory-level annotation or external instruction rewriting is involved. This task offers a reliable benchmark for evaluating atomic web interaction capabilities in a static, visually grounded setting.

**Noisy Multi Step Web Task**    To further enhance the original OpenWebVoyager (He et al., 2024b) dataset, we incorporate **Knowledge-driven Chain-of-Thought (CoT) Reasoning** to improve the model's stepwise understanding and execution, details see Section 4.2. In addition, to simulate realistic interruptions during multi-step web interactions, we propose the **Noisy Multi-Step Web Task** by augmenting interaction trajectories from OpenWebVoyager. Specifically, for each sample in our **Popup Close** dataset, a popup window is injected at a specific step (e.g., step $t$) of an existing task trajectory.

This modification introduces a prerequisite interaction: the agent must first detect and dismiss the popup before resuming progress toward the original task goal. By explicitly modeling such interruptive UI elements, this task formulation captures a more realistic web interaction paradigm in which user flows are frequently obstructed. It also provides a challenging benchmark for evaluating agents' robustness to UI-level noise and their capacity for error recovery.

## D   TRAINING METHODOLOGIES AND EVALUATION PROTOCOL

### D.1   TRAINING IMPLEMENTATION AND STRATEGIES

We employ a multi-phase Imitation Learning strategy to train our model on **Web-CogDataset**, utilizing Qwen2.5-VL-7B (Bai et al., 2025) as the base model. Each phase is aligned with a distinct layer of the Web-CogKnowledge Framework: (1) the first knowledge content learning focuses on acquiring Factual Knowledge and Conceptual Knowledge, enabling the model to interpret web content and semantics; (2) the second cognitive process emphasizes Procedural Knowledge, training the model to plan and execute multi-step web interactions. To accommodate the increased complexity of the final phase, which involves multi-image inputs and extended reasoning, we configure training with a maximum sequence length of 8K and a batch size of 1 with gradient accumulation of 16 steps. All experiments are conducted on a cluster of 8 x NVIDIA A800 80GB GPUs.

We could feed these SFT datasets into models with different training strategies. In this section, We investigate how they influence the model's performance.

1. Curriculum Learning Strategy: We fine-tune the model following a curriculum learning paradigm.

2. Mixed Multi-task Learning: We directly mix different tasks and apply SFT.

Through comparative analysis, we find that: Under the same task scenario, models trained via curriculum learning conduct multiple rounds of exploration continuously and do not cease exploration prematurely. In contrast, when retrieving task-related data, models trained via mixed training terminate the search within a limited number of attempts if they fail to find results.

As shown in Figure 7, when faced with the user's instruction "Open the most helpful 5 - star reviews of Alpine Ridge", models trained via mixed training deviate from the domain of task - specific information retrieval and autonomously generate an instruction to "switch to forums for information".

Based on these observations, we conjecture: Curriculum learning trains models incrementally from simple to complex tasks, which enables them to develop a fundamental understanding of the problem structure. They continue to carry out multiple rounds of exploration to ensure comprehensive comprehension and accurate processing, thus avoiding the premature termination of exploration. In contrast, mixed training models may be trained on multiple tasks simultaneously. The potential interference among different tasks can make models easily influenced by irrelevant tasks when processing a specific one, thereby undermining their ability to focus on task - specific information retrieval and processing.

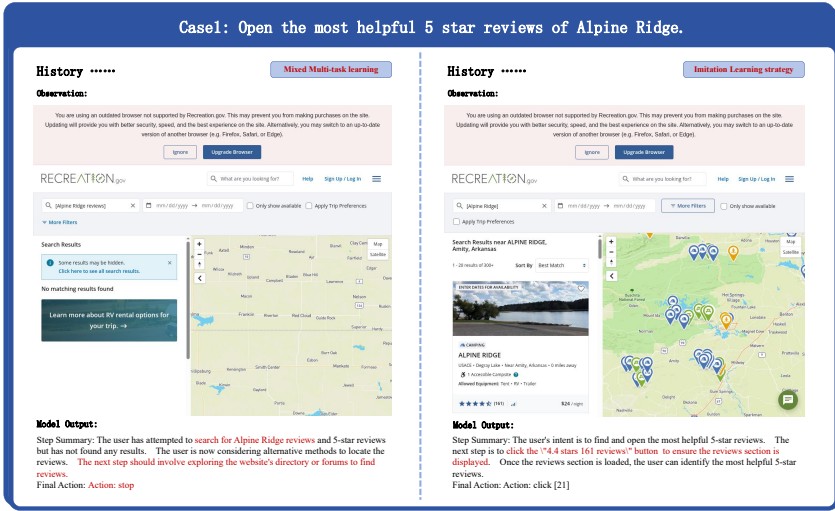

Figure 7: Comparison between mixed and curriculum strategies.

## D.2    LVM-BASED EVALUATION PROTOCOL

For open-ended generation tasks in Web-CogBench (i.e., Element Understanding, WebPage Understanding, and User Intent Prediction), we utilize a high-capability LVM (GPT-4o) as an automated evaluator. The evaluation process involves a strict comparison between the **Candidate Model's Answer** and the verified **Ground Truth** based on image provided in the dataset.

To ensure fine-grained assessment, we decompose the evaluation into specific cognitive dimensions rather than relying on a single holistic score:

- **Element Understanding:** Assessed on *Appearance* (visual fidelity), *Position* (structural context), and *Function* (interaction purpose).

- **WebPage Understanding:** Assessed on *Structure & Layout*, *Key Element Analysis*, and *Summary Coherence*.

- **User Intent Prediction:** Assessed on *Evidence Alignment* (visual cue detection), *Intent Accuracy*, and *Reasoning Quality*.

We employ a rigorous **1-5 Likert Scale** and normalized to 0–100 for scoring. A score of 5 denotes that the candidate "fully and accurately captures all relevant information present in the Ground Truth," while lower scores reflect varying degrees of omissions or inaccuracies. The evaluator outputs a structured JSON object containing both integer scores and text justifications for each dimension, ensuring the traceability of the results.

For instance, the *Element Understanding* task is assessed on:

**System Instruction:** You are a meticulous and impartial AI evaluator for a web UI understanding benchmark. Your task is to assess the quality of a candidate model's answer by comparing it strictly against a ground truth and provided image reference.
Your evaluation must be based *exclusively* on the information provided in the "Ground Truth Answer" and "Image".
Evaluate the candidate answer on three specific aspects: Appearance, Position, and Function.
**[Ground Truth Answer]**
{ground_truth}
—
**[Candidate Model's Answer]**
{model_answer}
—
**Evaluation Criteria & Scoring:**

- **Score 1:** Completely incorrect or missing.

- **Score 2:** Mostly incorrect, with a minor element of truth.

- **Score 3:** Partially correct, but misses significant details mentioned in the ground truth.

- **Score 4:** Mostly correct, with only minor inaccuracies or omissions compared to the ground truth.

- **Score 5:** Fully and accurately captures all relevant information present in the ground truth.

Your response MUST be a single, valid JSON object, adhering to the following structure. Do not add any text before or after the JSON object.
{
"appearance_score": ¡integer_score¿,
"appearance_justification": "¡Your brief justification... referencing the ground truth¿",
"position_score": ¡integer_score¿,
"position_justification": "¡Your brief justification... referencing the ground truth¿",
"function_score": ¡integer_score¿,
"function_justification": "¡Your brief justification... referencing the ground truth¿",
"overall_score": ¡A final holistic integer score from 1 to 5¿,
"overall_justification": "¡A final summary of the model's performance¿"
}

## D.3 EVALUATOR RELIABILITY ANALYSIS

To mitigate potential biases from a single evaluator, we employed **multiple distinct LVMs** (including GPT-4o, Claude Sonnet 4, and Gemini 2.5 Pro) to conduct a rigorous inter-rater reliability analysis. We calculate the **"Within-1-Point Agreement"**, defined as the percentage of instances where scores assigned by different LVM judges differ by no more than 1 point.

As shown in Table 14, the high agreement rates across different models confirm that our evaluation criteria are robust and model-agnostic. Furthermore, the strong correlation with Human Proxy Analysis suggests that our *Ground-Truth Anchored* protocol effectively aligns automated judgment with human evaluation standards.

Table 14: Inter-Rater Reliability Analysis of LVM Judge.

| Task | Within-1-Point Agreement | Human Proxy Analysis |
|---|---|---|
| Element Understanding | 98.7% | 96.7% |
| WebPage Understanding | 97.0% | 95.4% |
| User Intent Prediction | 96.0% | 94.4% |

## E EXTENDED QUALITATIVE ANALYSIS

To provide deeper insights into Web-CogReasoner's capabilities, we present a two-part qualitative analysis. First, we examine the **evolution of cognitive abilities** across training stages to validate our curriculum. Second, we present a **comparative case study** on a complex real-world task to demonstrate how our model overcomes knowledge blind spots that trap baseline models.

## E.1 EVOLUTION OF COGNITIVE ABILITIES ACROSS STAGES

Beyond the quantitative improvements shown in our ablation study, a qualitative analysis of the agent's behavior at each stage offers deeper insights into how our curriculum shapes its cognitive abilities. We examine the agents' performance on a representative task: "Find and add a laptop under $1000 to the shopping cart on an e-commerce website."

**Base Model (Qwen2.5-VL-7B)**   Without any specialized training, the base model struggles to formulate a coherent plan. Its reasoning is often generic and untethered from the specific UI. It might correctly identify a "search bar" but fails to execute a meaningful action, or hallucinates actions that are not possible. For instance, its thought process might be: *"I should search for a laptop,"* but its action is an ungrounded 'click "Categories"' because it lacks the procedural knowledge to connect intent to a multi-step sequence of actions.

**Stage 1 Agent (+ Factual Knowledge)**   After training on Factual Knowledge, the agent's perceptual abilities are significantly enhanced. It can now accurately identify and label key elements with their correct attributes. Its thought process becomes grounded in the facts of the page: *"I see a search bar [ID: 25] with the name 'Search products'. I see a button [ID: 28] with the name 'Search'."* However, it still struggles with planning. It understands "what" is on the page but not "why" or "how" to use it. It might correctly type "laptop" into the search bar but then get stuck, not understanding that the next logical step is to click the search button to submit the query.

**Stage 2 Agent (+ Conceptual Knowledge)**   With the addition of Conceptual Knowledge, the agent begins to understand the relationships between elements and their purpose. Its reasoning graduates from simple identification to semantic interpretation. The thought process now reflects this understanding: *"The search bar [ID: 25] is for inputting queries. The search button [ID: 28] is functionally linked to it and will trigger the search. This group of elements forms a 'search component'."* This allows it to reliably complete the search and navigate to the result page. However, on the result page, it may still struggle with complex procedural logic, such as applying a price filter.

**Full Model (Web-CogReasoner + Procedural Knowledge)**   The final agent, equipped with Procedural Knowledge, demonstrates goal-oriented planning and execution. It seamlessly translates the high-level task into a concrete action sequence. Its thought process is now a strategic plan: *"Goal: Add laptop under $1000 to cart. **Step 1:** Type 'laptop' into search bar [ID: 25]. **Step 2:** Click search button [ID: 28]. **Step 3:** On the results page, locate the 'Price Range' filter. **Step 4:** Input '1000' into the 'max price' field [ID: 57]. **Step 5:** Identify a suitable product from the filtered list and click its 'Add to Cart' button [ID: 83]."* This demonstrates a complete cognitive loop from perception and understanding to successful action, validating the necessity of the final procedural training stage.

### E.2   COMPARATIVE CASE STUDY ON REAL-WORLD TASKS

To validate the necessity of foundational knowledge in handling complex real-world scenarios, we compare the **Base Model** with **Web-CogReasoner** on a specific Amazon task.

**Task:** "Find a gaming desktop with Windows 11 Home and 1TB disk."

**Base Model Failure: The Knowledge Blind Spot**   Lacking explicit knowledge of page layout and element functions, the Base Model literally "sees" the pixels but "misses" the affordance.

- **Observation:** The model sees the search results but fails to recognize the sidebar filters as the mechanism to refine the query.
- **Error:** It misinterprets the page state, assuming a re-search is necessary. It enters a infinite loop of repeatedly clicking the search button.
- **Action:** `click [1470] (Search Button)` → **Stuck**.

**Web-CogReasoner Success: Knowledge-Driven Grounding**   Leveraging learned Factual and Conceptual knowledge, our agent explicitly identifies the functional role of UI components.

- **Reasoning:** The agent identifies the sidebar as a filter section. It conceptually maps the user's "1TB" requirement to the specific filter element, predicting that clicking it will narrow the results without leaving the page.
- **Action:** `click [95] (Filter Link "1 TB")` → **Success**.

This comparison highlights that success in complex tasks is not just about planning (Procedural), but requires strictly accurate recognition of element functions (Factual/Conceptual) to ground those plans.

