# OpenReview forum: "Web-CogReasoner: Towards Multimodal Knowledge-Induced Cognitive Reasoning for Web Agents"
_ICLR.cc/2026/Conference — ICLR 2026 Poster_

### Official Review · Reviewer_YsZ8 · 2025-10-25

**Soundness:** 3
**Presentation:** 4
**Contribution:** 3
**Rating:** 6
**Confidence:** 4

**Summary:**

The paper Web-CogReasoner: Towards Knowledge-Induced Cognitive Reasoning for Web Agents proposes a novel cognitive learning framework for web agents, inspired by Bloom’s taxonomy of human learning. The authors argue that effective web agents must acquire three types of knowledge—Factual, Conceptual, and Procedural—to perform reasoning and exploration tasks on the web. They introduce three key components: Web-CogKnowledge Framework, Web-CogDataset, and Web-CogBench, enabling a systematic curriculum that teaches agents to memorize, understand, and explore. The resulting model, Web-CogReasoner, integrates knowledge-driven chain-of-thought (CoT) reasoning, achieving 84.4% accuracy on Web-CogBench, outperforming Gemini 2.5 Pro, Claude Sonnet 4, and prior open-source agents on reasoning-intensive tasks

**Strengths:**

This work presents a conceptually rich and methodologically rigorous framework that advances the frontier of web-agent cognition. By grounding web reasoning in educational psychology (Bloom’s taxonomy), the authors provide a human-analogous paradigm for hierarchical learning—progressing from factual perception to procedural reasoning. The proposed dataset and benchmark cover 14 real-world websites with fine-grained multimodal tasks, yielding both scalability and interpretability. The experiments are comprehensive and ablation-supported, demonstrating clear contributions from each knowledge layer. The structured chain-of-thought (CoT) design substantially improves cross-task generalization and reasoning interpretability, and the authors provide transparent reporting and reproducibility details. Overall, the work represents a significant step toward cognitively inspired, knowledge-grounded web agents.

**Weaknesses:**

Despite its novelty, the study has several limitations. The model diversity is narrow—experiments primarily rely on Qwen 2.5 VL-7B, without testing multiple scales or architectures (e.g., Llama, Claude Opus, GPT-5), which constrains external validity. The framework’s dependence on supervised imitation learning raises concerns about long-horizon exploration, and its performance gap in cross-web generalization (≈10%) suggests limited adaptability beyond curated domains. Additionally, the theoretical connection between Bloom’s taxonomy and CoT reasoning is conceptually interesting but lacks formal grounding or cognitive validation. The framework also does not address multimodal robustness under dynamic webpages or visual perturbations, and scalability to open-world, continuously evolving web environments remains uncertain.

**Questions:**

1. How would Web-CogReasoner perform if trained with reinforcement learning or self-play instead of pure imitation, to strengthen procedural adaptability?

2. Could the authors include more diverse model families and scales (e.g., 7B–70B, open- and closed-source) to validate generality across architectures?

3. Have the authors explored integrating multimodal cognition—such as combining visual-textual reasoning with real-time interaction on dynamic or partially observable webpages?

4. To what extent is the Bloom’s-inspired hierarchy empirically necessary—would a non-hierarchical or single-stage variant achieve comparable performance?

5. Can the framework be extended toward lifelong or self-directed learning, enabling agents to autonomously acquire new procedural knowledge from unstructured online experiences?

---

> ### Author Response · Authors · 2025-11-24
> **Response to Reviewer YsZ8 Q1 & Q2 & Q3 (Part 1/2)**
>
> We sincerely appreciate your thoughtful review, constructive suggestions, and the time invested in evaluating our work. We address your concerns as follows.
>
> **Q1): How would Web-CogReasoner perform if trained with reinforcement learning or self-play instead of pure imitation, to strengthen procedural adaptability?**
>
> Thanks for your insight. We recognize the promise of RL and self-play, but they are orthogonal to our main contribution. Our work centers on a complete cognitive-data-evaluation framework, not on optimization algorithms.  Even within an Imitation Learning paradigm, our approach demonstrates robust interactive capabilities through three key mechanisms:
>
> 1. **Reasoning Activation via KCoT:**
>
>    As shown in `Table R1`, unlike generic Chain-of-Thought, our KCoT is intrinsic to our framework. It explicitly bridges our knowledge and action, boosting success rates from **25.35%** (w/o KCoT) to **42.9%** (w/ KCoT). This proves the framework successfully operationalizes static knowledge into dynamic decision-making.
>
>    > **Table R1. Ablation results on WebVoyager. S1/2/3 correspond to Factual/Conceptual/Procedural knowledge. KCoT is Knowledge-Driven CoT**
>    >
>    > | Model                   |  Amazon   | Cambridge | Coursera  |  GitHub   | **Overall** |
>    > | :---------------------- | :-------: | :-------: | :-------: | :-------: | :---------: |
>    > | **S1+S2+S3 (w/o KCoT)** |  19.51%   |  51.16%   |  26.19%   |  23.80%   |   25.35%    |
>    > | **S1+S2+S3 (w/ KCoT)**  | **31.7%** | **55.8%** | **54.8%** | **29.3%** |  **42.9%**  |
>
> 2. **Resilience via Noisy Training:**
>
>    Our training goes beyond simple cloning by integrating Noisy Multi-Step Tasks (e.g., interruptions). This forces the agent to learn error recovery and state adaptation, effectively instilling the resilience required for exploration.
>
> 3. **Real-World Validation:**
>
>    Superior performance on WebVoyager and Online Mind2Web (live, unscripted environments) empirically confirms the agent's capability to plan and adapt dynamically in real-world scenarios.
>
> ---
>
> **Q2): Could the authors include more diverse model families and scales (e.g., 7B–70B, open- and closed-source) to validate generality across architectures?**
>
> We appreciate the suggestion. We agree that scaling up model size would likely enhance reasoning performance under our framework. Our error analysis indicates that many failures stem from hallucination issues inherent to 7B-level models, such as misinterpreting date selectors in booking tasks. However, due to computational constraints, we are currently unable to train and evaluate models at larger parameter scales in the short term.
>
> > **Task:** "Book a flight for next Friday (May 31)."
> >
> > **Failure:** The 7B agent correctly plans: *"Today is May 24, I must select May 31."* However, it executes: click [Date_30].
> >
> > **Insight:** The framework successfully guided the correct procedural planning, but the 7B backbone failed at atomic precision. Such execution-level errors are specific to model capacity and would likely be resolved by scaling to larger architectures (e.g., 72B).
>
> ---
>
> **Q3): Have the authors explored integrating multimodal cognition—such as combining visual-textual reasoning with real-time interaction on dynamic or partially observable webpages?**
>
> We model web interaction as a **POMDP**, explicitly fusing visual perception (screenshots) with textual structure (AxTree) to reason about partially observable states. To ensure robustness in real-time dynamics, we train on **Noisy Multi-Step Tasks** (e.g., handling popups/interruptions), compelling the agent to combine multimodal cues for error recovery and state adaptation. This capability is rigorously validated on **live benchmarks** (WebVoyager, Online Mind2Web), where the agent successfully navigates unscripted, changing environments in real-time.

---

> ### Author Response · Authors · 2025-11-24
> **Response to Reviewer YsZ8 Q4 & Q5 (Part 2/2)**
>
> **Q4): To what extent is the Bloom’s-inspired hierarchy empirically necessary—would a non-hierarchical or single-stage variant achieve comparable performance?**
>
> Following your advice, as shown in `Table R2 & R3`, we conducted a complete ablation study isolating each stage (S1/2/3) as well as their combinations, demonstrating the importance of hierarchical results within our framework. We have updated the paper and added these details to the Appendix. We summarize the key findings as follows:
>
> 1. **Low-level Knowledge is a prerequisite for higher-level reasoning**:
>
>    As shown in `Table R2`, foundation knowledge scaffolds higher-level reasoning. Adding Factual training (S1) significantly boosts Conceptual Understanding (**S12 vs. S2: 75.5 vs. 68.03**) and Procedural Exploration (**S13 vs. S3: 82.31 vs. 78.00**).
>
> 2. **Integration is Critical for Complex Tasks**:
>
>    As shown in `Table R3`, single-stage models fail on real-world navigation (hovering ~12-13%). Combining stages unlocks capabilities; e.g., S13 nearly doubles the success rate of S3 alone (**23.47% vs. 13.14%**), proving online complex tasks cannot be solved without the support of knowledge at all levels.
>
> 3. **Activation via KCoT**:
>
>    While S123 builds the latent representation, KCoT is essential to explicitly link this knowledge for decision-making. It bridges the gap between having knowledge and using it, boosting WebVoyager performance from **25.35%** (w/o KCoT) to **42.9%** (w/ KCoT).
>
> > **Table R2. Ablation results on Web-CogBench. S1/2/3 correspond to Factual/Conceptual/Procedural knowledge.**
> >
> > | Model          |  Memory  | Understanding | Exploration |  Overall  |
> > | :------------- | :------: | :-----------: | :---------: | :-------: |
> > | **Base model** |   67.6   |     61.0      |    77.9     |   69.81   |
> > | **S1**         |   85.5   |     64.2      |    60.1     |   70.65   |
> > | **S2**         |  59.88   |     68.03     |    60.00    |   61.96   |
> > | **S3**         |  52.82   |     46.40     |    78.00    |   60.66   |
> > | **S1+S2**      |   88.1   |   **75.5**    |    65.8     |   76.59   |
> > | **S1+S3**      |  85.11   |     53.53     |    82.31    |   76.17   |
> > | **S2+S3**      |  64.87   |     69.74     |    81.41    |   72.29   |
> > | **S1+S2+S3**   | **90.8** |     74.1      |  **85.0**   | **84.45** |
>
> >**Table R3. Ablation results on WebVoyager. S1/2/3 correspond to Factual/Conceptual/Procedural knowledge.**
> >
> >| Model                   |  Amazon   | Cambridge | Coursera  |  GitHub   | **Overall** |
> >| :---------------------- | :-------: | :-------: | :-------: | :-------: | :---------: |
> >| **S1**                  |  12.19%   |  25.58%   |  14.28%   |   7.14%   |   12.67%    |
> >| **S2**                  |   9.75%   |  23.25%   |   9.52%   |  14.28%   |   12.20%    |
> >| **S3**                  |  17.07%   |  11.62%   |  16.66%   |  14.28%   |   13.14%    |
> >| **S1+S2**               |   9.75%   |  16.27%   |  26.19%   |  16.66%   |   14.55%    |
> >| **S1+S3**               |  29.26%   |  34.88%   |  28.57%   |  16.66%   |   23.47%    |
> >| **S2+S3**               |  21.95%   |  46.51%   |  21.42%   |  14.28%   |   21.59%    |
> >| **S1+S2+S3 (w/o KCoT)** |  19.51%   |  51.16%   |  26.19%   |  23.80%   |   25.35%    |
> >| **S1+S2+S3 (w/ KCoT)**  | **31.7%** | **55.8%** | **54.8%** | **29.3%** |  **42.9%**  |
>
> ---
>
> **Q5): Can the framework be extended toward lifelong or self-directed learning, enabling agents to autonomously acquire new procedural knowledge from unstructured online experiences?**
>
> Thanks for your insight. Our framework serves as an ideal **"Cognitive Base"** for lifelong learning. This separation provides a structured mechanism to convert unstructured experiences into new skills:
>
> 1. **General-to-Specific Knowledge Instantiation:**
>
>    Factual and Conceptual knowledge (e.g., recognizing "Login" patterns or "Filter" logic) are highly transferable across the web. In a new environment, the agent can leverage this pre-existing Cognitive Base to parse unfamiliar interfaces zero-shot. It can identify functional affordances (S1 & S2) to hypothesize potential action sequences, thereby jumpstarting the acquisition of site-specific Procedural knowledge (S3) without starting from scratch.
> 2. **Self-Supervised Procedural Synthesis:**
>
>    The Conceptual Knowledge layer (Understanding) serves as an internal "Verifier." When the agent explores unstructuredly, it can use its understanding capabilities to assess the semantic outcome of its actions (e.g., "Did the page state change match my intent?"). This allows the agent to autonomously filter successful trajectories from noisy interactions, effectively synthesizing new Procedural training data from its own experiences to update its policy continuously.
> ---
> In summary, we sincerely appreciate the reviewers’ constructive feedback. **We provide a “Guideline for Reviewers” in the appendix A.6, where all major revisions are highlighted with color to specific reviewer**

---

### Official Review · Reviewer_PZyc · 2025-11-01

**Soundness:** 3
**Presentation:** 3
**Contribution:** 2
**Rating:** 6
**Confidence:** 3

**Summary:**

This paper proposes Web-CogReasoner, a knowledge-induced cognitive reasoning framework for web agents. The key idea is to structure agent reasoning with a knowledge-driven chain-of-thought (K-CoT) aligned to three layers—factual, conceptual, and procedural—and to train a Qwen2.5-VL-7B–based agent via staged supervision on a new dataset (Web-CogDataset) collected from real websites. A companion benchmark (Web-CogBench) evaluates three capability dimensions—memory, understanding, and exploration—that map to the knowledge layers. Experiments report strong performance on Web-CogBench and competitive results on established web-agent benchmarks (e.g., VisualWebBench, WebVoyager, Mind2Web / Online Mind2Web), with ablations showing complementary gains from all three knowledge layers. These design choices are well-motivated by trends in web-agent evaluation and environment standardization.

**Strengths:**

**Clear, systematic framework**: The use of Bloom’s taxonomy to structure the agent’s learning and reasoning is conceptually coherent. While knowledge-layered modeling is not entirely novel, applying it concretely to web agents with CoT alignment is a valuable contribution.

**Well-structured data and evaluation system**: The pairing of Web-CogDataset and Web-CogBench, mapped by knowledge layers and ability dimensions, allows end-to-end evaluation from knowledge acquisition to interactive behavior.

**Reproducibility awareness and engineering detail**: The paper reports training setups, hardware, sequence lengths, and strategy choices, and includes responsible research statements. This transparency helps reproducibility and fosters trust.

**Weaknesses:**

**Website sampling bias**: The chosen 14 sites may disproportionately lean toward e-commerce or financial domains. This could bias the model’s strengths toward those styles; broader site diversity (e.g., social media, education, news, forums) would strengthen generalization.

**Lack of strong baseline comparisons with GPT-level models**: Without direct comparisons to large generalist models (e.g. GPT-4, GPT-5), it is hard to assess how far this approach is from “state-of-the-art general reasoning agents.”

**Questions:**

**Reliance on pseudo-labels and their reliability**: Some annotations are generated by Qwen-VL-72B; the paper should report the error rates, whether they performed human double-blind verification, or whether cross-model annotation consistency was checked (e.g., compare labels from Claude / Gemini).

**Judge bias / evaluator correlation**: The use of an LVM Judge for scoring may be correlated with the model training biases or data domain. Do authors provide inter-judge agreement or bootstrap confidence intervals, especially on the Understanding / Exploring tasks?

**Limited demonstration of active exploration / online feedback**: While procedural knowledge and exploration are core aims, the paper heavily leans toward structured CoT and imitation learning. More experiments incorporating online decision-making (e.g. RL fine-tuning, exploration during inference) would strengthen claims about interactive capability.

---

> ### Author Response · Authors · 2025-11-24
> **Response to Reviewer PZyc W1 (Part 1/3)**
>
> We sincerely appreciate your thoughtful review, constructive suggestions, and the time invested in evaluating our work. We address your concerns as follows.
>
> **W1): Website sampling bias: The chosen 14 sites may disproportionately lean toward e-commerce or financial domains. This could bias the model’s strengths toward those styles; broader site diversity (e.g., social media, education, news, forums) would strengthen generalization.**
>
> We appreciate the suggestion regarding domain diversity. We wish to clarify that the 14 websites represent only the self-collected subset of our data, used for high-depth interaction mining. The full Web-CogDataset is a strategic hybrid of this self-collected data and augmented open-source corpora, ensuring both depth and broad generalization.
>
> 1. **Balance within Self-Collected Data:**
>
>    As illustrated in `Table R1 & Figure 5`, even within the 14 self-collected sites, we strictly maintained category balance across E-commerce, Finance, Developer Tools, and Social & Media, rather than focusing solely on financial/commercial domains. This ensures that the high-quality, deep-interaction data is structurally diverse.
>
>    > **Table R1. Website Categories in Web-CogDataset (Manual Collection)**
>    >
>    > | Category            | Proportion |                Websites                |
>    > | ------------------- | :--------: | :------------------------------------: |
>    > | **E-commerce**      |   28.6%    |      Amazon, Ebay, 12306, Airbnb       |
>    > | **Social & Media**  |   28.6%    |  Zhihu, Weibo, Bilibili, Apple Music   |
>    > | **Finance**         |   21.4%    |     Binance, Coinglass, Eastmoney      |
>    > | **Developer Tools** |   21.4%    | Github, Stack Overflow, Stack Exchange |
>
> 2. **Broad Generalization via Open-Source Augmentation:**
>
>    To achieve open-domain coverage (including News, Education, and Forums), we incorporated and enhanced large-scale datasets such as MultiUI [1], Mind2Web [2], and OpenWebVoyager [3]. Notably, MultiUI is derived from FineWeb (Common Crawl), which encompasses a massive variety of general-purpose webpages. By integrating these diverse sources, our dataset avoids domain overfitting and ensures the model generalizes effectively to the "wild" web.

---

> ### Author Response · Authors · 2025-11-24
> **Response to Reviewer PZyc W2 & Q1 (Part 2/3)**
>
> **W2): Lack of strong baseline comparisons with GPT-level models: Without direct comparisons to large generalist models (e.g. GPT-4, GPT-5), it is hard to assess how far this approach is from “state-of-the-art general reasoning agents.”**
>
> Thanks for your advice. We respectfully note that our evaluation already benchmarks against the most advanced generalist models (`Claude Sonnet 4` and `Gemini 2.5 Pro`). To further address your concern for GPT-series comparisons, as shown in `Table R2`, we have incorporated `GPT-4o` into our evaluation on WebVoyager [4].
>
> > **Table R2 (Part 1/2). Task success rates on WebVoyager with GPT-4o**
> >
> > | Agent                        | Allrecipes |  Amazon   |   Apple   |   ArXiv   |  GitHub   |  Booking  |   ESPN    | Coursera  |
> > | :--------------------------- | :--------: | :-------: | :-------: | :-------: | :-------: | :-------: | :-------: | :-------: |
> > | Claude Sonnet 4              |   26.7%    |   87.8%   |   48.8%   |   69.8%   |   68.3%   |   2.3%    |   45.5%   |   83.3%   |
> > | Gemini 2.5 Pro               |   60.0%    |   63.4%   |   62.8%   |   67.4%   |   68.3%   |   9.1%    |   56.8%   |   73.8%   |
> > | **GPT-4o**                   |   54.3%    |   51.7%   |   58.3%   |   62.5%   |   62.5%   |   13.9%   |   44.0%   |   70.1%   |
> > | Qwen2.5-VL-7B                |    0.0%    |   0.0%    |   0.0%    |   4.7%    |   0.0%    |   0.0%    |   0.0%    |   2.3%    |
> > | OpenWebVoyager<sub>IL</sub>  |   17.8%    |   12.2%   |   20.9%   |   14.0%   |   14.6%   |   9.1%    |   9.1%    |   31.0%   |
> > | OpenWebVoyager<sub>Max</sub> |   22.2%    |   29.3%   |   32.6%   |   20.9%   |   26.8%   | **11.4%** |   11.4%   |   42.9%   |
> > | **Web-CogReasoner (Ours)**   | **26.7%**  | **31.7%** | **32.6%** | **34.9%** | **29.3%** |   2.3%    | **15.9%** | **54.8%** |
> >
> > **Table R2 (Part 2/2). Task success rates on WebVoyager with GPT-4o**
> >
> > | Agent                        | BBC News  | Cambridge Dict | Google Flights | Google Map | Huggingface | Wolfram Alpha | **Overall** |
> > | :--------------------------- | :-------- | :------------- | :------------- | :--------- | :---------- | :------------ | :---------: |
> > | Claude Sonnet 4              | 23.8%     | 37.2%          | 4.8%           | 80.5%      | 48.8%       | 82.6%         |    47.7%    |
> > | Gemini 2.5 Pro               | 52.3%     | 76.7%          | 4.8%           | 75.6%      | 58.1%       | 82.6%         |    54.9%    |
> > | **GPT-4o**                   | 54.8%     | 78.2%          | 8.6%           | 76.9%      | 42.6%       | 75.2%         |    52.6%    |
> > | Qwen2.5-VL-7B                | 0.0%      | 11.6%          | 0.0%           | 2.4%       | 7.0%        | 2.2%          |    2.2%     |
> > | OpenWebVoyager<sub>IL</sub>  | 9.5%      | 37.2%          | 9.5%           | 22.0%      | 20.9%       | 26.1%         |    18.1%    |
> > | OpenWebVoyager<sub>Max</sub> | 14.3%     | 34.9%          | **21.4%**      | 29.3%      | 32.6%       | 37.0%         |    26.2%    |
> > | **Web-CogReasoner (Ours)**   | **14.3%** | **55.8%**      | 9.5%           | **39.0%**  | **37.2%**   | **39.1%**     |  **30.2%**  |
>
> ---
>
> **Q1): Reliance on pseudo-labels and their reliability: Some annotations are generated by Qwen-VL-72B; the paper should report the error rates, whether they performed human double-blind verification, or whether cross-model annotation consistency was checked (e.g., compare labels from Claude / Gemini).**
>
> Thank you for pointing out the issue. As shown in `Table R3`, we validated our annotations via double-blind human verification and cross-model checks (`GPT-4o`, `Gemini 2.5 Pro` and `Claude Sonnet 4`.), demonstrating > 96% accuracy. This reliability stems from our Ground-Truth Guided pipeline: the teacher model is explicitly conditioned on recorded interaction logs and AxTrees. It functions as an interpreter of verified facts rather than a generator, ensuring minimal hallucination. We have added these details to the Appendix.
>
> > **Table R3: Reliability Check of Web-CogDataset Annotations**
> >
> > | Annotation Task        | Human Verification (Accuracy) | Cross-Model Consistency (GPT/Claude/Gemini) |
> > | ---------------------- | :---------------------------: | :-----------------------------------------: |
> > | Element Attribute      |             99.2%             |                    98.5%                    |
> > | Page Change Prediction |             97.5%             |                    96.8%                    |
> > | Sub-element Prediction |             96.8%             |                    95.4%                    |
> > | **Average**            |           **97.8%**           |                  **96.9%**                  |

---

> ### Author Response · Authors · 2025-11-24
> **Response to Reviewer PZyc Q2 & Q3 (Part 3/3)**
>
> **Q2): Judge bias / evaluator correlation: The use of an LVM Judge for scoring may be correlated with the model training biases or data domain. Do authors provide inter-judge agreement or bootstrap confidence intervals, especially on the Understanding / Exploring tasks?**
>
> Thanks for the suggestion. For the LVM-Judge in Web-CogBench, each task is scored on a multi-dimensional 1–5 Likert scale and normalized to 0–100. To mitigate bias, we utilize multiple LVMs. As shown in `Table R4`, validation on a randomly sampled 1/3 subset confirms robust consistency with human proxies. External benchmarks (e.g., WebVoyager, online Mind2Web) strictly follow their original protocols. We will update the paper and added these details to the Appendix.
>
> > **Table R4: Inter-Rater Reliability Analysis.** "Within-1-Point Agreement" indicates the percentage of scores that differ by no more than 1 point between judges.
> >
> > | Task                   | Average Within-1-Point Agreement | Human Proxy Analysis |
> > | ---------------------- | :------------------------------: | :------------------: |
> > | Element Understanding  |              98.7%               |        96.7%         |
> > | WebPage Understanding  |              97.0%               |        95.4%         |
> > | User Intent Prediction |              96.0%               |        94.4%         |
>
> ---
>
> **Q3): Limited demonstration of active exploration / online feedback: While procedural knowledge and exploration are core aims, the paper heavily leans toward structured CoT and imitation learning. More experiments incorporating online decision-making (e.g. RL fine-tuning, exploration during inference) would strengthen claims about interactive capability.**
>
> We acknowledge the potential of RL, yet emphasize that it is orthogonal to our core contribution: the systematic Web-CogKnowledge Framework. This framework establishes a rigorous closed loop—*Bloom’s Taxonomy → Web-CogDataset → Knowledge-Driven CoT → Capability Verification*. Even within an Imitation Learning paradigm, our approach demonstrates robust interactive capabilities through three key mechanisms:
>
> 1. **Reasoning Activation via KCoT:**
>
>    As shown in `Table R5`, unlike generic Chain-of-Thought, our KCoT is intrinsic to our framework. It explicitly bridges our knowledge and action, boosting success rates from **25.35%** (w/o KCoT) to **42.9%** (w/ KCoT). This proves the framework successfully operationalizes static knowledge into dynamic decision-making.
>
>    > **Table R5. Ablation results on WebVoyager. S1/2/3 correspond to Factual/Conceptual/Procedural knowledge.**
>    >
>    > | Model                   |  Amazon   | Cambridge | Coursera  |  GitHub   | **Overall** |
>    > | :---------------------- | :-------: | :-------: | :-------: | :-------: | :---------: |
>    > | **S1**                  |  12.19%   |  25.58%   |  14.28%   |   7.14%   |   12.67%    |
>    > | **S2**                  |   9.75%   |  23.25%   |   9.52%   |  14.28%   |   12.20%    |
>    > | **S3**                  |  17.07%   |  11.62%   |  16.66%   |  14.28%   |   13.14%    |
>    > | **S1+S2**               |   9.75%   |  16.27%   |  26.19%   |  16.66%   |   14.55%    |
>    > | **S1+S3**               |  29.26%   |  34.88%   |  28.57%   |  16.66%   |   23.47%    |
>    > | **S2+S3**               |  21.95%   |  46.51%   |  21.42%   |  14.28%   |   21.59%    |
>    > | **S1+S2+S3 (w/o KCoT)** |  19.51%   |  51.16%   |  26.19%   |  23.80%   |   25.35%    |
>    > | **S1+S2+S3 (w/ KCoT)**  | **31.7%** | **55.8%** | **54.8%** | **29.3%** |  **42.9%**  |
>
> 2. **Resilience via Noisy Training:**
>
>    Our training goes beyond simple cloning by integrating Noisy Multi-Step Tasks (e.g., interruptions). This forces the agent to learn error recovery and state adaptation, effectively instilling the resilience required for exploration.
>
> 3. **Real-World Validation:**
>
>    Superior performance on WebVoyager and Online Mind2Web (live, unscripted environments) empirically confirms the agent's capability to plan and adapt dynamically in real-world scenarios.
>
> ---
>
> [1] Liu, Junpeng, et al. "Harnessing webpage uis for text-rich visual understanding." 2024.
>
> [2] Cheng, Kanzhi, et al. "Seeclick: Harnessing gui grounding for advanced visual gui agents." 2024.
>
> [3] He, Hongliang, et al. "Openwebvoyager: Building multimodal web agents via iterative real-world exploration, feedback and optimization." 2025.
>
> [4] He, Hongliang, et al. "Webvoyager: Building an end-to-end web agent with large multimodal models." 2024.
>
> ---
> In summary, we sincerely appreciate the reviewers’ constructive feedback. **We provide a “Guideline for Reviewers” in the appendix A.6, where all major revisions are highlighted with color to clearly indicate how each change corresponds to specific reviewer comments, enabling quick and convenient inspection.**

---

### Official Review · Reviewer_HEcV · 2025-11-01

**Soundness:** 3
**Presentation:** 3
**Contribution:** 2
**Rating:** 2
**Confidence:** 4

**Summary:**

This paper proposes a Web-CogKnowledge framework inspired by Bloom's taxonomy, where web navigation knowledge is categorized into three levels and a two-phase training methodology is followed in order to enhance the cognitive capabilities of web agents. Based on this framework, the authors construct Web-CogDataset and use this dataset to create Web-CogBench. Furthermore, the authors train a Web-CogReasoner model and conduct evaluations against several popular models on multiple benchmarks.

Overall, this is an interesting paper attempting to frame web agent learning processes from a cognitive perspective following Bloom's taxonomy. However, all core ingredients are well-explored by related work under different names, resulting in insufficient technical novelty.

**Strengths:**

- Strong motivation and interesting idea to frame the problem using Bloom's taxonomy, attempting to model the learning process as three progressive stages (Memorizing, Understanding, Exploring) that map to different knowledge types.

- The dataset is well-structured following the proposed framework, covering 12 fine-grained tasks across factual, conceptual, and procedural knowledge, which could be a valuable contribution to the community for training models and conducting evaluations.

**Weaknesses:**

### Major

- The three proposed stages are essentially different names for well-established concepts like visual grounding, VQA, and action prediction that are widely used in current web agents, just reframed under cognitive terminology.

- To strength the argument of the proposed framework, the ablation study should demonstrate performance with and without each individual stage, rather than progressively adding stages one by one, to fully investigate the independent impact of each component in the proposed framework.

- Evaluations should be more comprehensive, missing some key benchmarks: static grounding tasks such as ScreenSpot series and dynamic environments such as WebArena and VisualWebArena that better reflect real-world web interaction scenarios.

- Web-CogBench remains static in nature and doesn't capture the dynamic, interactive aspects of real web environments where agents must adapt to changing page states and handle unexpected scenarios

### Minor

Line #343: 'via upervised fine-tuning' should be 'via supervised fine-tuning'

**Questions:**

- Regarding the POMDP formulation – how is the Markovian property maintained when previous action history is included in the 'trajectory history' section? Doesn't this violate the memoryless assumption?
- In Web-CogDataset, how do you define the single-step and multiple-step web tasks? What's the source?
- Can more details about the LVM Judge be disclosed?
- How is accuracy measured for open-ended responses in next page prediction?

---

> ### Author Response · Authors · 2025-11-24
> **Response to Reviewer HEcV W1 (Part 1/4)**
>
> We sincerely appreciate your thoughtful review, constructive suggestions, and the time invested in evaluating our work. We address your concerns as follows.
>
> **W1) The three proposed stages are essentially different names for well-established concepts like visual grounding, VQA, and action prediction that are widely used in current web agents, just reframed under cognitive terminology.**
>
> We respectfully argue that our contribution lies not in inventing new interaction formats, but in the systematic categorization and hierarchical induction of knowledge—a paradigm shift from *functional capability* to *cognitive development*. We treat them as vehicles to instantiate different levels of Bloom's Taxonomy rather than isolated functional skills. We address your concerns from the following three points.
>
> 1. **Most Tasks Are Newly Designed for Knowledge Acquisition**:
>
>    Our tasks are constructed to support a specific knowledge-acquisition pipeline. As shown in Figure 2, **9 out of 12** tasks are newly collected and rigorously defined based on cognitive principles, distinguishing them from traditional WebVQA tasks (isolated functional tasks), thus establishing a structured "Factual-Conceptual-Procedural" trajectory where each layer builds upon the previous one. For example in **conceptual knowledge** tasks:
>
>    * Element Understanding grounds the previously acquired **factual knowledge**, e.g., element attributes, relations, affordances.
>    * WebPage Understanding attempts to extend this to **procedural knowledge** with layout analysis, element understanding, and page intent inference.
>
> 2. **Hierarchical Dependency is the Core**:
>
>    Prior work treats VQA and grounding as standalone functional tasks. In contrast, we prioritize hierarchical knowledge accumulation aligned with Bloom's Taxonomy, using our reconstructed hierarchical tasks. For instance, while both Factual and Conceptual layers employ QA formats, they differ strictly in cognitive complexity: mastering the higher-level conceptual knowledge functionally depends on the prerequisite factual knowledge. Our ablation study in `Table R1 & R2` confirms this.
>
> 3. **Our framework establishes a cognitive paradigm**:
>
>    Inspired by Bloom's Taxonomy, our "cognitive labels" are not just post-hoc classifications but functional mechanisms. On the one hand, we inject knowledge into the web agent through meticulously designed tasks and hierarchical training. On the other hand, through Knowledge-driven Chain-of-Thought (KCoT), we compel the model to explicitly activate Factual, Conceptual, and Procedural knowledge before taking real online tasks, transforming these theoretical categories into practical reasoning steps.

---

> ### Author Response · Authors · 2025-11-24
> **Response to Reviewer HEcV W2 (Part 2/4)**
>
> **W2) To strength the argument of the proposed framework, the ablation study should demonstrate performance with and without each individual stage**
>
> Following your advice, as shown in `Table R1` and `Table R2`, we conducted a complete ablation study isolating each stage (S1/2/3) as well as their combinations. We will update the paper and added these details to the Appendix. We summarize the key findings as follows:
>
> 1. **Low-level Knowledge is a prerequisite for higher-level reasoning**:
>
>    As shown in `Table R1`, foundation knowledge scaffolds higher-level reasoning. Adding Factual training (S1) significantly boosts Conceptual Understanding (**S12 vs. S2: 75.5 vs. 68.03**) and Procedural Exploration (**S13 vs. S3: 82.31 vs. 78.00**).
>
> 2. **Integration is Critical for Complex Tasks**:
>
>    As shown in `Table R2`, single-stage models fail on real-world navigation (hovering ~12-13%). Combining stages unlocks capabilities; e.g., S13 nearly doubles the success rate of S3 alone (**23.47% vs. 13.14%**), proving online complex tasks cannot be solved without the support of knowledge at all levels.
>
> 3. **Activation via KCoT**:
>
>    While S123 builds the latent representation, KCoT is essential to explicitly link this knowledge for decision-making. It bridges the gap between having knowledge and using it, boosting WebVoyager performance from **25.35%** (w/o KCoT) to **42.9%** (w/ KCoT).
>
> > **Table R1. Ablation results on Web-CogBench. S1/2/3 correspond to Factual/Conceptual/Procedural knowledge.**
> >
> > | Model          |  Memory  | Understanding | Exploration |  Overall  |
> > | :------------- | :------: | :-----------: | :---------: | :-------: |
> > | **Base model** |   67.6   |     61.0      |    77.9     |   69.81   |
> > | **S1**         |   85.5   |     64.2      |    60.1     |   70.65   |
> > | **S2**         |  59.88   |     68.03     |    60.00    |   61.96   |
> > | **S3**         |  52.82   |     46.40     |    78.00    |   60.66   |
> > | **S1+S2**      |   88.1   |   **75.5**    |    65.8     |   76.59   |
> > | **S1+S3**      |  85.11   |     53.53     |    82.31    |   76.17   |
> > | **S2+S3**      |  64.87   |     69.74     |    81.41    |   72.29   |
> > | **S1+S2+S3**   | **90.8** |     74.1      |  **85.0**   | **84.45** |
>
> >**Table R2. Ablation results on WebVoyager. S1/2/3 correspond to Factual/Conceptual/Procedural knowledge.**
> >
> >| Model                   |  Amazon   | Cambridge | Coursera  |  GitHub   | **Overall** |
> >| :---------------------- | :-------: | :-------: | :-------: | :-------: | :---------: |
> >| **S1**                  |  12.19%   |  25.58%   |  14.28%   |   7.14%   |   12.67%    |
> >| **S2**                  |   9.75%   |  23.25%   |   9.52%   |  14.28%   |   12.20%    |
> >| **S3**                  |  17.07%   |  11.62%   |  16.66%   |  14.28%   |   13.14%    |
> >| **S1+S2**               |   9.75%   |  16.27%   |  26.19%   |  16.66%   |   14.55%    |
> >| **S1+S3**               |  29.26%   |  34.88%   |  28.57%   |  16.66%   |   23.47%    |
> >| **S2+S3**               |  21.95%   |  46.51%   |  21.42%   |  14.28%   |   21.59%    |
> >| **S1+S2+S3 (w/o KCoT)** |  19.51%   |  51.16%   |  26.19%   |  23.80%   |   25.35%    |
> >| **S1+S2+S3 (w/ KCoT)**  | **31.7%** | **55.8%** | **54.8%** | **29.3%** |  **42.9%**  |

---

> ### Author Response · Authors · 2025-11-24
> **Response to Reviewer HEcV W3 & W4 & Q1 (Part 3/4)**
>
> **W3): Evaluations should be more comprehensive, missing some key benchmarks that better reflect real-world web interaction scenarios.**
>
> Our current loop: VisualWebBench (fundamental visual ability), Web-CogBench (diagnostic), Online-Mind2Web, and WebVoyager (online task evaluation) already forms a closed, coherent set of evaluations that directly validates the capabilities taught by our unified cognitive training framework.
>
> 1. **Dynamic Environments (Live vs. Simulated)**:
>
>    Regarding WebArena [1]/VisualWebArena [2], we respectfully argue that our chosen benchmarks, WebVoyager and Online Mind2Web, better reflect real-world scenarios.
>
>    - **Reflect real capabilities:** WebArena [1] relies on simulated environments. In contrast, our benchmarks involve live, real-time interactions with actual websites, capturing the dynamic updates, latency, and complexity of the open web.
>
>    - **Robustness to Noise:** Real-world interaction is rarely perfect. Our framework is specifically trained on noisy trajectories (e.g., handling popups and unexpected interruptions). Testing in live online environments (rather than controlled simulations) provides the most authentic assessment of this noise-handling capability and error recovery.
>
> 2. **Static Grounding (VisualWebBench as a Strong Representative)**:
>
>    While we did not evaluate on ScreenSpot [3], we utilized VisualWebBench [4], a comprehensive benchmark that specifically targets static grounding capabilities (Element Grounding, Action Grounding). As shown in `Table R3`, our method achieves SOTA performance (e.g., **96.4%** on Element Grounding, **88.4%** on Action Grounding), effectively validating the model's precise grounding ability without the need for redundant static datasets.
>
> >**Table R3: Grounding Task Performance on VisualWebBench[4].**
> >
> >| Model                      | Element Ground | Action Prediction | Action Ground |   Avg    |
> >| :------------------------- | :------------: | :---------------: | :-----------: | :------: |
> >| **Claude Sonnet 4**        |      81.1      |       96.1        |   **96.3**    |   91.2   |
> >| **Gemini 2.5 Pro**         |      91.8      |     **96.8**      |     90.3      |   93.0   |
> >| **Qwen2.5-VL-7B**          |      77.5      |       86.8        |     68.0      |   77.4   |
> >| **UI-TARs-7B-SFT**         |      91.8      |       91.8        |     85.4      |   89.7   |
> >| **Web-CogReasoner (Ours)** |    **96.4**    |       96.1        |     88.4      | **93.6** |
>
> ---
>
> **W4): Web-CogBench remains static in nature and doesn't capture the dynamic, interactive aspects of real web environments where agents must adapt to changing page states and handle unexpected scenarios**
>
> Web-CogBench is intentionally not designed to simulate dynamic web interaction. Its purpose is to isolate and measure the fine-grained web knowledge required by an agent, serving as a diagnostic tool rather than an interaction environment.
>
> 1. **Direct Alignment with the Cognitive Framework.**
>
>    Web-CogBench evaluates the exact competencies defined in our framework and provides interpretable diagnostics that complement end-to-end dynamic evaluations. Real-world interaction is already covered by Online Mind2Web and WebVoyager, while Web-CogBench focuses on systematic problem localization.
>
> 2. **A Complementary Role in the Full Pipeline.**
>
>    Our workflow forms a closed loop—*Bloom’s Taxonomy → Web-CogDataset → knowledge-driven CoT → capability verification*. Web-CogBench provides the principle-driven knowledge assessment that anchors this pipeline.
>
> ---
>
> **W5): Line #343: 'via upervised fine-tuning' should be 'via supervised fine-tuning'**
>
> Thank you for pointing this out. We have corrected the typo from *“via upervised fine-tuning”* to *“via supervised fine-tuning”* in the revised version.
>
> ---
>
> **Q1): Regarding the POMDP formulation – how is the Markovian property maintained when previous action history is included in the 'trajectory history' section? Doesn't this violate the memoryless assumption?**
>
> Including trajectory history does not violate the Markov property. The Markovian assumption applies to the environment's transition dynamics $P(s_{t+1} \mid s_t, a_t)$, ensuring the next state depends only on the current state and action. It does not constrain the agent's policy. In a POMDP setting, since the true state is partially observable, the policy $\pi(a_t \mid h_t)$ must rely on history to estimate the underlying belief state for optimal decision-making. This is a standard formulation consistent with Markovian environments. We provide relevant work defined in the same manner for reference (e.g., WebVoyager [5], AgentGym [6], Agent Q [7]).

---

> ### Author Response · Authors · 2025-11-24
> **Response to Reviewer HEcV Q2 & Q3 & Q4 (Part 4/4)**
>
> **Q2): In Web-CogDataset, how do you define the single-step and multiple-step web tasks? What's the source?**
>
> Their definitions and data sources are as follows. More details about task definations is in Appendix.
>
> * **Single-Step Task**: A one-page screenshot, a natural-language instruction, and a set of candidate clickable elements are provided as input. The model must select the correct element as the output. Source: Multi-UI [8].
> * **Noisy Multi-Step Task**: Real exploration trajectories (image, AXTree, action) from OpenWebVoyager [9], augmented with KCoT and injected popup events to introduce noise.
>
> ---
>
> **Q3): Can more details about the LVM Judge be disclosed?**
>
> Thanks for the suggestion. For the LVM-Judge in Web-CogBench, each task is scored on a multi-dimensional 1–5 Likert scale and normalized to 0–100. To mitigate bias, we utilize multiple LVMs. As shown in `Table R4`, validation on a randomly sampled 1/3 subset confirms robust consistency with human proxies. External benchmarks strictly follow their original protocols. We will update the paper and added these details to the Appendix.
>
> > **Table R4: Inter-Rater Reliability Analysis. "Within-1-Point Agreement" indicates the percentage of scores that differ by no more than 1 point between judges.**
> >
> > | Task                   | Average Within-1-Point Agreement | Human Proxy Analysis |
> > | ---------------------- | :------------------------------: | :------------------: |
> > | Element Understanding  |              98.7%               |        96.7%         |
> > | WebPage Understanding  |              97.0%               |        95.4%         |
> > | User Intent Prediction |              96.0%               |        94.4%         |
>
> ---
>
> **Q4): How is accuracy measured for open-ended responses in next page prediction?**
>
> Thanks for your question. The training and evaluation formats differ. Training combines multiple-choice and open-ended data at a 1:4 ratio to improve both alignment and generalization. However, Web-CogBench uses multiple-choice questions for this task to ensure precise Accuracy measurement.
>
> ---
>
> [1] Zhou, Shuyan, et al. "Webarena: A realistic web environment for building autonomous agents." 2023.
>
> [2] Koh, Jing Yu, et al. "Visualwebarena: Evaluating multimodal agents on realistic visual web tasks." 2024.
>
> [3] Cheng, Kanzhi, et al. "Seeclick: Harnessing gui grounding for advanced visual gui agents." 2024.
>
> [4] Liu, Junpeng, et al. "Visualwebbench: How far have multimodal llms evolved in web page understanding and grounding?." 2024.
>
> [5] He, Hongliang, et al. "Webvoyager: Building an end-to-end web agent with large multimodal models." 2024.
>
> [6] Xi, Zhiheng, et al. "Agentgym: Evolving large language model-based agents across diverse environments." 2024.
>
> [7] Putta, Pranav, et al. "Agent q: Advanced reasoning and learning for autonomous ai agents." 2024.
>
> [8] Liu, Junpeng, et al. "Harnessing webpage uis for text-rich visual understanding." 2024.
>
> [9] He, Hongliang, et al. "Openwebvoyager: Building multimodal web agents via iterative real-world exploration, feedback and optimization." 2025.
>
> ---
> In summary, we sincerely appreciate the reviewers’ constructive feedback. **We provide a “Guideline for Reviewers” in the appendix A.6, where all major revisions are highlighted with color to clearly indicate how each change corresponds to specific reviewer comments, enabling quick and convenient inspection.**

---

### Official Review · Reviewer_ri9L · 2025-11-01

**Soundness:** 2
**Presentation:** 2
**Contribution:** 3
**Rating:** 4
**Confidence:** 3

**Summary:**

The paper proposes Web-CogReasoner, a web agent trained with a two stage, knowledge first paradigm inspired by Bloom’s taxonomy: first learn Factual, Conceptual, and Procedural web knowledge, then exercise cognitive processes for reasoning and action. The authors build Web-CogDataset from 14 real sites with 12 fine-grained tasks, and introduce Web-CogBench to evaluate “Memorizing, Understanding, Exploring” capabilities aligned to those knowledge types. A  Web-CogReasoner is further trained with Qwen2.5-VL-7B as base model and  achieves strong performance on benchmarks including Web-CogBench and VisualWebBench.

**Strengths:**

The evaluation of Web-CogBench is aligned to the training data construction framework.  Web-CogBench measures “Memorizing, Understanding, Exploring,” directly mirroring the training objectives and enabling targeted diagnosis of capabilities.

The combination of screenshots with the accessibility tree leverages both visual cues and structured semantics of web pages.

**Weaknesses:**

Authors have placed much emphasis on the three aspects of “factual, conceptual, and procedural learning,” claiming that “higher-order reasoning is built on solid perceptual and conceptual foundations.” It would be great to see supporting experiments.

It would be helpful to provide more details around the selection of “memorizing, understanding, and exploring” abilities as the tasks for Web-CogBench. Some questions that may be good to answer are: how well do these aspects cover the actual tasks performed by web agents. Are these necessary conditions for a strong web agent.

It would be good to know the reason behind choosing webvoyager and online-mind2web as the only two end to end web agent benchmarks, as they both rely on live websites and environments, which can bring noise to the experiment. In particular, it would be good to know the reason behind choosing online multimodal mind2web over multimodal mind2web. And it would be great to provide more details around experiment setup, e.g. is human or LLM judge used as evaluation. If Web-CogReasoner is set to take the role of a deep research style agent, then live environment with benchmarks such as GAIA could be a good choice.

**Questions:**

Listed above.

---

> ### Author Response · Authors · 2025-11-24
> **Response to Reviewer ri9L W1 (Part 1/3)**
>
> We sincerely appreciate your thoughtful review, constructive suggestions, and the time invested in evaluating our work. We address your concerns as follows.
>
> **W1): Lack experiments for "higher-order reasoning is built on solid perceptual and conceptual foundations".**
>
> Thank you for this highly insightful suggestion. In response, as shown in `Table R1 & R2`, we have incorporated comprehensive quantitative ablation studies and qualitative case analyses to support our claim. We wiil update the paper and added these details to the Appendix.
>
> 1. **quantitative ablation studies:**
>
>    * **Low-level Knowledge is a prerequisite for higher-level reasoning**:
>
>      As shown in `Table R1`, foundation knowledge scaffolds higher-level reasoning. Adding Factual training (S1) significantly boosts Conceptual Understanding (**S12 vs. S2: 75.5 vs. 68.03**) and Procedural Exploration (**S13 vs. S3: 82.31 vs. 78.00**).
>
>    * **Integration is Critical for Complex Tasks**:
>
>      As shown in `Table R2`, single-stage models fail on real-world navigation (hovering ~12-13%). Combining stages unlocks capabilities; e.g., S13 nearly doubles the success rate of S3 alone (**23.47% vs. 13.14%**), proving online complex tasks cannot be solved without the support of knowledge at all levels.
>
>    * **Activation via KCoT**:
>
>      While S123 builds the latent representation, KCoT is essential to explicitly link this knowledge for decision-making. It bridges the gap between having knowledge and using it, boosting WebVoyager performance from **25.35%** (w/o KCoT) to **42.9%** (w/ KCoT).
>
>    >**Table R1. Ablation results on Web-CogBench. S1/2/3 correspond to Factual/Conceptual/Procedural knowledge.**
>    >
>    >| Model          |  Memory  | Understanding | Exploration |  Overall  |
>    >| :------------- | :------: | :-----------: | :---------: | :-------: |
>    >| **Base model** |   67.6   |     61.0      |    77.9     |   69.81   |
>    >| **S1**         |   85.5   |     64.2      |    60.1     |   70.65   |
>    >| **S2**         |  59.88   |     68.03     |    60.00    |   61.96   |
>    >| **S3**         |  52.82   |     46.40     |    78.00    |   60.66   |
>    >| **S1+S2**      |   88.1   |   **75.5**    |    65.8     |   76.59   |
>    >| **S1+S3**      |  85.11   |     53.53     |    82.31    |   76.17   |
>    >| **S2+S3**      |  64.87   |     69.74     |    81.41    |   72.29   |
>    >| **S1+S2+S3**   | **90.8** |     74.1      |  **85.0**   | **84.45** |
>    >
>    >**Table R2. Ablation results on WebVoyager. S1/2/3 correspond to Factual/Conceptual/Procedural knowledge.**
>    >
>    >| Model                   |  Amazon   | Cambridge | Coursera  |  GitHub   | **Overall** |
>    >| :---------------------- | :-------: | :-------: | :-------: | :-------: | :---------: |
>    >| **S1**                  |  12.19%   |  25.58%   |  14.28%   |   7.14%   |   12.67%    |
>    >| **S2**                  |   9.75%   |  23.25%   |   9.52%   |  14.28%   |   12.20%    |
>    >| **S3**                  |  17.07%   |  11.62%   |  16.66%   |  14.28%   |   13.14%    |
>    >| **S1+S2**               |   9.75%   |  16.27%   |  26.19%   |  16.66%   |   14.55%    |
>    >| **S1+S3**               |  29.26%   |  34.88%   |  28.57%   |  16.66%   |   23.47%    |
>    >| **S2+S3**               |  21.95%   |  46.51%   |  21.42%   |  14.28%   |   21.59%    |
>    >| **S1+S2+S3 (w/o KCoT)** |  19.51%   |  51.16%   |  26.19%   |  23.80%   |   25.35%    |
>    >| **S1+S2+S3 (w/ KCoT)**  | **31.7%** | **55.8%** | **54.8%** | **29.3%** |  **42.9%**  |
>
> 2. **qualitative case analyses:**
>
>    To validate the necessity of foundational knowledge, we compare the **Base Model** (Qwen2.5-VL-7B) with **Web-CogReasoner** on a real-world task. The Base Model's failure is not an execution error but a knowledge deficit, it literally "sees" the pixels but "misses" the affordance. Web-CogReasoner succeeds because high-level reasoning is strictly grounded in the accurate recognition of element functions.
>
>    > **Case Task:** "Find a gaming desktop with Windows 11 Home and 1TB disk" on Amazon.
>    >
>    > - **Base Model (Failure): Knowledge Blind Spot**
>    >   Lacking knowledge of page layout and element function, the model fails to recognize the sidebar filters. It misinterprets the page state, assuming a re-search is necessary, and enters a logical dead loop of repeatedly clicking the search button.
>    >
>    >   > **Action:** click [1470] (Search Button) → **Stuck**
>    >
>    > - **Web-CogReasoner (Success): Knowledge-Driven Grounding**
>    >   Leveraging learned Factual and Conceptual knowledge, our agent explicitly identifies the functional role of the sidebar. It correctly maps the user's "1TB" requirement to the specific filter element and predicts that clicking it will narrow the results.
>    >
>    >   > **Action:** click [95] (Filter Link "1 TB") → **Success**

---

> ### Author Response · Authors · 2025-11-24
> **Response to Reviewer ri9L W2 (Part 2/3)**
>
> **W2): It would be helpful to provide more details around the selection of “memorizing, understanding, and exploring” abilities as the tasks for Web-CogBench. Some questions that may be good to answer are: how well do these aspects cover the actual tasks performed by web agents. Are these necessary conditions for a strong web agent.**
>
> We selected Memorizing, Understanding, and Exploring because they map directly to the fundamental cognitive loop required for any autonomous agent: **Perception → Comprehension → Decision-making**.
>
> 1. **Necessity (Hierarchical Dependency):**
>    These dimensions form a strict prerequisite chain. Without Memorizing (identifying attributes), the agent is blind to the UI; without Understanding (comprehending function), it cannot interpret what it sees; without Exploring (strategic execution), it cannot act. Our ablation studies (`Tables R1 & R2`) confirm that removing any layer leads to systematic failure, proving they are necessary conditions for a robust web agent.
>
> 2. **Coverage (Real-World Case Study):**
>
>    These three dimensions cover the atomic operations of web interaction. To demonstrate this, we analyze a typical task: *"Find a 24-hour parking lot and check reviews on Google Maps."* The execution relies entirely on the interplay of these three specific abilities:
>
>    > **Step 1: Search Query**
>    >
>    > *   *Action:* Identify the search box and input "Brooklyn Bridge parking lot."
>    > *   *Cognitive Mapping:*
>    >     *   **Memorizing (Attributes & State):** Accurately identifies the element's attributes (e.g., Role: `combobox`, Name: `Search Google Maps`) and recognizes the current page state (initial map view).
>    >     *   **Understanding (Function & Semantics):** Comprehends the element's function (accepts text input for location queries) and its contextual relationship to the page (primary navigation tool).
>    >     *   **Exploring (Goal-Oriented Action):** Plans and executes the specific action (typing the target query) to drive the task forward.
>    >
>    > **Step 2: Selection**
>    >
>    > *   *Action:* Scan the results list and click "LAZ Parking – The 1 Hotel."
>    > *   *Cognitive Mapping:*
>    >     *   **Memorizing:** Identifies the attributes of the new list elements (Role: `link`, Text: `24 hours`) generated by the page state change.
>    >     *   **Understanding:** Interprets the semantics of "24 hours" as matching the user's criteria and understands the list item's function (clicking leads to details).
>    >     *   **Exploring:** Strategically selects the correct target from multiple candidates to fulfill the specific sub-goal.
>    >
>    > **Step 3: Information Retrieval**
>    >
>    > *   *Action:* Locate and click the "Reviews" tab.
>    > *   *Cognitive Mapping:*
>    >     *   **Memorizing:** Recalls the specific attribute of the "Reviews" tab (Role: `tab`).
>    >     *   **Understanding:** Comprehends the deep semantics that "Reviews" contains the requested user feedback.
>    >     *   **Exploring:** Executes the final navigational step to complete the multi-step user goal.

---

> ### Author Response · Authors · 2025-11-24
> **Response to Reviewer ri9L W3 & W4 (Part 3/3)**
>
> **W3): It would be good to know the reason behind choosing webvoyager and online-mind2web as the only two end to end web agent benchmarks, as they both rely on live websites and environments, which can bring noise to the experiment. In particular, it would be good to know the reason behind choosing online multimodal mind2web over multimodal mind2web**
>
> We intentionally selected WebVoyager and Online Multimodal Mind2Web to prioritize real-world cognitive robustness. Our goal is to verify that the agent’s success stems from internalized knowledge reasoning rather than merely overfitting to fixed trajectories in static environments.
>
> 1.  **Validation of Cognitive Reasoning over Trajectory Fitting:**
>
>     Static environments often allow models to overfit to fixed interaction patterns. In contrast, the dynamic nature of live websites—with changing layouts and content—compels the agent to actively apply Factual and Conceptual knowledge to analyze the current state and infer the correct action. This proves that the agent truly "understands" the web, rather than merely fitting training trajectories.
> 2.  **Authenticity of Dynamic Noise (Live vs. Static):**
>
>     Unlike static benchmark, live environments introduce realistic challenges such as pop-ups and unexpected interruptions. This authentic noise is essential to verify the effectiveness of our "Noisy Multi-Step" training, assessing the agent's ability to handle uncertainty and recover from errors in ways that "clean" simulations cannot.
> 3.  **Technical Necessity:**
>
>     We prioritized the Online version over static Multimodal Mind2Web because the latter lacks Accessibility Tree (AxTree) metadata. Since our reasoning framework relies on AxTree for grounding, the static dataset is technically incompatible with our input requirements.
>
> **Note on Experimental Rigor:** To mitigate irrelevant environmental noise (e.g., network instability or latency) and ensure a fair assessment of capability, we conducted each task three times and reported the averaged results.
>
> ---
>
> **W4): It would be great to provide more details around experiment setup, e.g. is human or LLM judge used as evaluation.**
>
> Thanks for the suggestion. For the LVM-Judge in Web-CogBench, each task is scored on a multi-dimensional 1–5 Likert scale and normalized to 0–100. To mitigate bias, we utilize multiple LVMs. As shown in `Table R3`, validation on a randomly sampled 1/3 subset confirms robust consistency with human proxies. External benchmarks (e.g., WebVoyager, online Mind2Web) strictly follow their original protocols. We will update the paper and added these details to the Appendix.
>
> > **Table R3: Inter-Rater Reliability Analysis.** "Within-1-Point Agreement" indicates the percentage of scores that differ by no more than 1 point between judges.
> >
> > | Task                   | Average Within-1-Point Agreement | Human Proxy Analysis |
> > | ---------------------- | :------------------------------: | :------------------: |
> > | Element Understanding  |              98.7%               |        96.7%         |
> > | WebPage Understanding  |              97.0%               |        95.4%         |
> > | User Intent Prediction |              96.0%               |        94.4%         |
> ---
> In summary, we sincerely appreciate the reviewers’ constructive feedback. **We provide a “Guideline for Reviewers” in the appendix A.6, where all major revisions are highlighted with color to clearly indicate how each change corresponds to specific reviewer comments, enabling quick and convenient inspection.**

---

### Author Response · Authors · 2025-12-03
**Overall Response and Summary of Key Revisions**

We sincerely thank the reviewers for their constructive and insightful feedback, which significantly helped us strengthen both the experimental validation and the methodological rigor of **Web-CogReasoner**. We have carefully addressed each concern point-by-point and incorporated several key additions to support our claims and ensure robustness. We highlight four key additions that validate our findings:

1. **Extensive Ablation on Cognitive Hierarchy `(Tables R1 & R2 in reviewer ri9L)`:**

   To rigorously validate the necessity of our three-stage framework (Reviewers ri9L, HEcV, YsZ8), we conducted comprehensive ablation studies across two benchmarks (Web-CogBench and WebVoyager). We evaluated **8** distinct model variants (isolating S1, S2, S3, their combinations, and KCoT). The results confirm that foundational knowledge is a strict prerequisite: removing Factual/Conceptual stages causes a drastic drop in navigation success rates (from **23.47% to ~13%**), while our Knowledge-Driven CoT (KCoT) proves critical by bridging knowledge to action, boosting performance from **25.35% to 42.9%**.
2. **Benchmarking Rationale & Grounding Validation`(Table R3 in reviewer HEcV)`:**

   We clarified the necessity of using live environments (WebVoyager, Online Mind2Web) over static simulators to test robustness against real-world noise (Reviewers ri9L, HEcV). Furthermore, to validate our model's perceptual foundation, we reported results on VisualWebBench, where our method achieves exceptional performance (**93.6%** on Grounding Tasks), proving that our reasoning improvements are built on solid grounding capabilities.
3. **Comparison with Broad Generalist Models`(Tabel R2 in reviewer PZyc)`:**

   To address the suggestion for broader baseline comparisons (Reviewer PZyc), we supplemented our evaluation by adding **GPT-4o** on WebVoyager. This complements our existing comprehensive comparisons with top-tier models, including Claude Sonnet 4 and Gemini 2.5 Pro, providing a complete context to assess Web-CogReasoner's capabilities against the most capable closed-source agents.
4. **Data & Evaluation Reliability `(Tables R3 & R4 in reviewer PZyc)`:**

   We addressed concerns regarding potential bias in the LVM-Judge and dataset annotations (Reviewers ri9L, HEcV, PZyc). We conducted inter-rater agreement studies between our LVM Judge and human proxies, confirming **>96%** consistency. Additionally, double-blind human verification and cross-model checks (using GPT, Claude, and Gemini) on our dataset labels confirmed an accuracy of **>97%**, ensuring high data quality.

We again thank the reviewers for the opportunity to strengthen our work and for recognizing the contribution of our knowledge-induced cognitive framework to the web agent community.

---

> ### Author Response · Authors · 2025-12-03
> **Summary of Consolidated Reviewer Feedback**
>
> Below, we provide a consolidated summary of the strengths and key concerns raised, along with our corresponding responses. The summary highlights overlapping points. Detailed responses are in individual replies.
>
> **Strengths**
>
> 1. **Novel Cognitive Framework (HEcV, PZyc, YsZ8)**
>
>    Structuring learning into a hierarchical "Factual-Conceptual-Procedural" paradigm is recognized as conceptually coherent, rigorous, and offers a fresh perspective vs. purely functional approaches.
>
> 2. **Comprehensive Dataset and Evaluation System (ri9L, HEcV, PZyc, YsZ8)**
>
>    The Web-CogDataset (12 tasks) and Web-CogBench are well-regarded. Reviewers appreciated how evaluation dimensions map to training objectives, enabling targeted diagnosis of agent capabilities.
>
> 3. **Strong Performance and Reproducibility (ri9L, YsZ8)**
>
>    Web-CogReasoner shows strong benchmark performance. Transparent reporting of setups and detailed ablation studies were highlighted as significant contributions to reproducibility.
>
> **Weaknesses and Responses**
>
> 1. **Novelty of the Cognitive Framework (Reviewer HEcV)**
>
>    Reviewer HEcV questioned the technical novelty, suggesting the three stages might just be "relabel existing concepts."
>
>    **Response:**
>
>    We clarified our contribution is not renaming formats, but establishing a systematic hierarchical cognitive dependency, a paradigm shift from *functional capability* to *cognitive development*. Typical reference: `Response to Reviewer HEcV W1 (Part 1/4)`
>
>    * **New Task Designs:** **9 out of 12** tasks in Web-CogDataset are newly designed for this hierarchy, distinct from traditional VQA.
>    * **Cognitive Dependency:** Our framework enforces a strict dependency, e.g., conceptual understanding functionally relies on factual memory.
>    * **Mechanism vs. Label:** Our "cognitive labels" are functional mechanisms instantiated via Knowledge-Driven Chain-of-Thought (KCoT), explicitly activating layers sequentially for complex problem-solving.
>
> 2. **Empirical Necessity of the Three-Stage Hierarchy (Reviewers ri9L, HEcV, YsZ8)**
>
>    Reviewers requested empirical evidence that multi-stage training is superior and that "higher-order reasoning is truly built on solid perceptual foundations."
>
>    **Response:**
>
>    We conducted comprehensive quantitative ablation studies isolating each stage (S1/S2/S3) and their combinations to demonstrate the full hierarchy's necessity. Typical reference: `Response to Reviewer ri9L W1 (Part 1/3)`
>
>    * **Quantitative Evidence:** As shown in `Table R1 & R2`, foundation knowledge scaffolds higher-level reasoning.
>        * **Low-Level Prerequisite:** Factual training (S1) significantly boosts Conceptual (**S12 vs. S2: 75.5 vs. 68.03**) and Procedural Exploration (**S13 vs. S3: 82.31 vs. 78.00**).
>        * **Integration is Critical:** Single-stage models fail on real-world navigation (~12-13%). Combining Factual/Procedural (S13) nearly doubles the success rate (**23.47% vs. 13.14%**), proving multi-level integration is necessary.
>        * **KCoT Essential:** KCoT explicitly links and uses latent knowledge, boosting WebVoyager performance from **25.35% to 42.9%**.
>    * **Qualitative Analysis**: Added case studies proving high-level reasoning is strictly grounded in accurate low-level recognition.
>
> 3. **Benchmark Selection and Comparison with SOTA Models (Reviewers ri9L, HEcV, PZyc, YsZ8)**
>
>    Concerns were raised regarding the choice of live vs. static benchmarks, dataset bias and the lack of comparison with GPT-4o.
>
>    **Response:**
>
>    We justified live environments as essential for testing robustness against real-world noise. Typical reference: `Response to Reviewer HEcV W3 (Part 3/4)`
>
>    * **Benchmark Rationale:** We selected real-time benchmarks specifically to test cognitive reasoning and generalization under real-world dynamic noise, preventing overfitting to static trajectories. Our SOTA performance on VisualWebBench (e.g., **93.6%** on Grounding Tasks) validates our static grounding capabilities.
>    * **Dataset and Baselines:** We clarified that our dataset is a strategic hybrid of depth-balanced self-collected data and broad-coverage open-source corpora to ensure generalization. Furthermore, we added a direct comparison with GPT-4o on WebVoyager, contextualizing our model's competitive performance against closed-source SOTA models.
>
> 4. **Reliability of Dataset Annotations and Evaluation Metrics (Reviewers ri9L, HEcV, PZyc)**
>
>    Questions were raised about potential bias in the LVM-Judge and the reliability of pseudo-labels.
>
>    **Response:**
>
>    We implemented rigorous verification protocols. Typical reference: `Response to Reviewer PZyc Q1 & Q2`
>
>    * **Data Quality Verification:** Double-blind human/cross-model checks confirmed dataset label accuracy of **>97%** `(Table R3)`.
>
>    * **Inter-Rater Reliability:** LVM Judge consistency check showed **>96%** agreement with human proxies ` (Table R4)`, ensuring Web-CogBench scores' fairness.

---

> ### Author Response · Authors · 2025-12-03
>
> Dear Area Chair and Reviewers,
>
> We sincerely appreciate the constructive feedback. Appendix A.6 provides a “Guideline for Reviewers,” where all major revisions are color-highlighted and explicitly mapped to individual comments to facilitate quick and clear verification.
>
> We believe the revised manuscript fully addresses the raised concerns and better demonstrates the contribution of our cognitive framework.
>
> Regards,
>
> Authors

---

### Meta-Review · Area_Chair_uo9x · 2026-01-17

**Summary:**

This paper proposes to decompose web agent development into three types of knowledge, based on Bloom's taxononmy: factual, conceptual, and procedural. The paper uses this taxonomy to develop both a staged training strategy for web agents as well as a prompting-based method, and the authors also release a dataset + benchmark to support evaluation of each knowledge type.

A primary concern with the work is its novelty compared to other web agent research: the taxonomy by itself is a nice inclusion but prior work also accomplish similar things to each of the three layers described here. The training strategy was similarly questioned, but the auhtors did a good job in the rebuttal of reaffirming the value of the stages as well as the prompting strategy.

Overall, I think the authors did a great job addressing the primary concern, and the paper seems like a valuable contribution to the emerging area of web agent research!

**Reviewer Concerns:**

Satisfactorily addressed:
- value of staged training / prompting; addressed by extensive ablations
- comparisons to strong LLMs
- agreement between LLM judge and human raters

Still outstanding:
- incremental novelty: the taxonomy is interesting, but the components have all been studied before
- limited coverage of websites, not clear how generalizable the benchmark is
- studied "exploration" but doesn't delve into RL as a training strategy

**Reviewer Scores:**

ri9L: 4>6
HEcV: unchanged at 2
PZyc: 6>8
YsZ8: unchanged at 6

---

### Decision · Program_Chairs · 2026-01-26

Accept (Poster)